# Satellite mapping reveals extensive industrial activity at sea

Fernando S. Paolo[1,6 ✉], David Kroodsma[1,6], Jennifer Raynor[2], Tim Hochberg[1], Pete Davis[1], Jesse Cleary[3], Luca Marsaglia[1], Sara Orofino[4], Christian Thomas[5] & Patrick Halpin[3]

The world's population increasingly relies on the ocean for food, energy production and global trade[1–3], yet human activities at sea are not well quantified[4,5]. We combine satellite imagery, vessel GPS data and deep-learning models to map industrial vessel activities and offshore energy infrastructure across the world's coastal waters from 2017 to 2021. We find that 72–76% of the world's industrial fishing vessels are not publicly tracked, with much of that fishing taking place around South Asia, Southeast Asia and Africa. We also find that 21–30% of transport and energy vessel activity is missing from public tracking systems. Globally, fishing decreased by 12 ± 1% at the onset of the COVID-19 pandemic in 2020 and had not recovered to pre-pandemic levels by 2021. By contrast, transport and energy vessel activities were relatively unaffected during the same period. Offshore wind is growing rapidly, with most wind turbines confined to small areas of the ocean but surpassing the number of oil structures in 2021. Our map of ocean industrialization reveals changes in some of the most extensive and economically important human activities at sea.

More than one billion people depend on the ocean for their primary source of food[1–3], with 260 million employed by global marine fisheries alone[6]. About 80% of all traded goods are shipped over the ocean[7] and nearly 30% of the world's oil is produced in offshore fields and distributed worldwide[8]. In addition to these established uses of the ocean, increases in offshore renewable energy, aquaculture and mining are rapidly emerging. All of this industrial machinery powers a 1.5–2.5 trillion dollar 'blue economy'[9,10] that is growing faster than the overall global economy[10] but is also causing rapid environmental decline. A third of fish stocks are operated beyond biologically sustainable levels[11] and an estimated 30–50% of critical marine habitats have been lost owing to human industrialization[12–14].

A lack of global observational data limits understanding of where and how the blue economy is expanding and how it is affecting developing nations and coastal communities[15–17]. On land, maps exist for almost every road[18], datasets are being developed for every human-made structure[19] and extractive industries such as forestry and agriculture are mapped globally at sub-kilometre scale and updated monthly[20,21]. In the ocean, however, many seagoing vessels do not broadcast their location or are not detected by public monitoring systems[22], and information on the development of offshore infrastructure and other industrial activities is often held private[23]. The result is that continuing human expansion into the ocean is poorly documented.

Current approaches for mapping human activity at sea have limitations. Some vessel-tracking systems, such as the vessel monitoring system (VMS) used in fishing, are proprietary, which limit the ability to map and compare across regions[5]. For public mapping of ships, the focus has been on the automatic identification system (AIS)[24], which broadcasts vessel coordinates to track vessel movements and support maritime safety; AIS data can also reveal vessel identities, owners and corporations, and fishing activities[5,25,26]. Not all vessels, however, are required to use AIS devices, as regulations vary by country, vessel size and activity[22]. Vessels engaged in illicit activities often turn off their AIS transponders or manipulate the locations they broadcast[27–29]. In recent years, for example, the largest cases of illegal fishing[28] and forced labour[30,31] were by fleets that mostly did not use AIS devices. Furthermore, large 'blind spots' along coastal waters emerge where satellite reception is poor[22] and AIS data received by terrestrial receptors can be restricted by national governments[32]. We refer to vessels that are not visible on publicly accessible AIS data as 'not publicly tracked'. This concept is also sometimes referred to as 'dark vessels'. Although the location of offshore fixed infrastructure should be more readily available than moving vessels, information on offshore development is often restricted for commercial or bureaucratic reasons[33], and large-scale assessments must aggregate several disparate data sources, which are often incomplete or outdated[34]. Vessel activity and ocean infrastructure are not captured well by existing methods, but satellite imagery and deep learning can improve the monitoring of human use of the ocean.

Here we present a detailed global map of major industrial activities at sea. To detect and characterize vessels and offshore infrastructure in coastal waters around the globe, we analysed 2 petabytes of satellite imagery spanning the years 2017–2021, with our analyses covering more than 15% of the ocean (Extended Data Fig. 1) in which more than 75% of industrial activity is concentrated (Methods). We designed and trained three deep convolutional neural networks to identify objects (>97% accuracy) and estimate their lengths ($R^2$ score

[1]Global Fishing Watch, Washington, DC, USA. [2]Forest and Wildlife Ecology Department, University of Wisconsin–Madison, Madison, WI, USA. [3]Marine Geospatial Ecology Lab, Nicholas School of the Environment, Duke University, Durham, NC, USA. [4]Bren School of Environmental Science and Management, University of California, Santa Barbara, Santa Barbara, CA, USA. [5]SkyTruth, Shepherdstown, WV, USA. [6]These authors contributed equally: Fernando S. Paolo, David Kroodsma. ✉e-mail: fernando@globalfishingwatch.org

of 0.84); to classify offshore infrastructure into oil, wind and other objects (>98% accuracy); and to classify vessels as fishing or non-fishing (>90% accuracy). Combined, we classified more than 67 million image tiles, including dual-polarization synthetic-aperture radar (SAR) imagery from Sentinel-1 (ref. 35) and optical (red, green, blue and near-infrared (NIR)) imagery from Sentinel-2 (ref. 36). The resolution of SAR allows us to capture most objects larger than 15 m (detection rate >70% for 25-m vessels and >90% for vessels 50 m and larger; Extended Data Fig. 2). We also analysed 53 billion vessel GPS positions from the AIS and matched them to the satellite detections to determine whether a detected vessel was publicly tracked.

## Fishing and non-fishing vessels

During 2017–2021, on average, about 63,300 vessel occurrences were detected at any given moment, roughly half (42–49%) of which were fishing vessels (based on 23.1 million vessel detections; Fig. 1). Notably, about three-quarters (72–76%) of globally mapped industrial fishing did not appear in public monitoring systems, compared with one-quarter (21–30%) for other vessel activities.

Vessel activity was widespread but also highly concentrated. Dividing our study area into 0.1° cells (about 11 km), we detected a vessel at least once in 84% of the cells covered by the satellites, yet half of all vessel activity was concentrated in less than 3% of the cells. Most vessel activity (86% of fishing and 75% of non-fishing) was focused in waters less than 200 m deep (Fig. 1), which constitute only 7% of the ocean. Activity is also unevenly distributed by continent, with approximately 67% of all vessel activity in Asia, followed by 12% in Europe, 7% in North America, 7% in Africa, 4% in South America and 2% in Australia (Fig. 1).

Our satellite mapping revealed high densities of vessel activity in large areas of the ocean that previously showed little to no vessel activity by public tracking systems (Fig. 2). Indonesia, South Asia, Southeast Asia and the northern and western coasts of Africa (Fig. 2 and Extended Data Figs. 3 and 4) all show substantial amounts of activity not publicly tracked.

By mapping vessels that fail to broadcast their location, we show far more accurately the global distribution of industrial fishing. AIS data alone, for example, wrongly suggest that Europe and Asia have comparable fishing activity, with other continents having less than one-fifth as much activity (Extended Data Table 1). Our global map, however, reveals that Asia dominates industrial fishing, accounting for 70% of all fishing vessel detections (Extended Data Fig. 5); nearly 30% of all mapped fishing vessels were concentrated in the exclusive economic zone (EEZ) of China alone. Similarly, AIS data suggest that European countries in the Mediterranean have more than ten times as many fishing hours in their EEZs as do African countries[5], but our mapping shows that detections of fishing vessels are fairly balanced between the northern and southern parts of the Mediterranean Sea (Figs. 1 and 2).

Our mapping can also reveal potential hotspots of illegal fishing activity. Previous work showed substantial illicit activity in the eastern waters of North Korea[28], but our global mapping shows that most of the undisclosed fishing actually occurred in the western part of the Korean Peninsula (Fig. 2). In fact, this location showed the highest density of fishing vessels in the world from 2017 to 2019, with about 40 vessels per 1,000 km². This previously unmapped activity peaked each year in May, during China's moratorium on fishing in their own waters (Extended Data Fig. 6), and activity abruptly fell by 85% during the COVID-19 pandemic when North Korea shut its borders. Numerous fishing vessels not publicly tracked were also detected inside many marine protected areas (MPAs). For example, two of the most iconic, biologically important and well-monitored MPAs in the world—the Galápagos Marine Reserve and the Great Barrier Reef Marine Park—showed, on average, more than 5 and 20 of these vessels per week, respectively (Extended Data Fig. 7).

The spatial resolution of our data, which is substantially higher than the most widely used global fishing products[37,38], also reveals detailed fishing strategies at the regional scale (Fig. 2 and Extended Data Fig. 3). The area between Tunisia and Sicily, for example, shows a mix of both publicly and not publicly tracked fishing vessels aggregating along ocean banks and the edges of seabed canyons, a signature characteristic of bottom trawling[39]. Similarly, off the coast of Bangladesh, in which almost no vessels are publicly tracked and no public maps of fishing exist, fishing vessels follow bathymetric contours and submarine canyons that radiate from the Ganges Delta.

Unlike fishing, most non-fishing vessels (largely transport and energy-related) broadcast their locations, with just about one-quarter missing from public monitoring systems. Asia had the largest concentration (65% of all detections) of transport and energy vessels, including most of the non-broadcasting ones (Fig. 1)—most of these vessels, however, were operating in areas with poor satellite AIS reception, so it is possible that many vessels broadcast their positions but were not trackable with global AIS tracking services. All other continents seemed to have relatively minor tracking discrepancies across transport and energy vessels, with less than 20% of these vessels not publicly trackable.

Our mapping also tracks changes in vessel activity over time (Fig. 3). Similar to a previous AIS-based analysis[5], our data show yearly cycles of fishing activity, with cycles inside China driven by the Chinese New Year and their voluntary fishing moratorium, and in the rest of the world by the New Year and associated holidays. But, owing to SAR-based detection, we can provide a more accurate assessment of trends, which reveals a global decrease in fishing activity of 12 ± 1%, coinciding with the pandemic. By stark contrast, transport and energy remained stable or even slightly increased over 2017–2021. Moreover, the impact of COVID-19 on fishing activity was much greater outside China (compared with 2018 and 2019), and transport and energy grew more in China than it did in the rest of the world.

## Fixed infrastructure

The number of offshore structures worldwide was around 28,000 by the end of 2021 (Fig. 4). Wind turbines and oil structures in notable wind-producing or oil-producing areas (Methods) constituted 48% and 38% of all ocean infrastructure, respectively; the remaining 14% was divided across wind turbines and oil structures outside major development areas, as well as piers, bridges, power lines, aquaculture and other human-made structures.

Most oil infrastructure is distributed among 13 major oil-producing areas (Fig. 4a). Excluding Lake Maracaibo in Venezuela, which is a lagoon, our mapping shows that the largest concentration of offshore oil infrastructure in the world is in the Gulf of Mexico. At the end of 2021, about a quarter of the global offshore oil infrastructure is accounted for by the USA (>2,200 oil structures), followed by Saudi Arabia (>770) and Indonesia (>670).

Offshore wind development has been mostly confined to northern Europe (52%) and China (45%) (Fig. 4a and Extended Data Fig. 8); however, there has been a shift in offshore energy development. The number of offshore oil structures has increased by about 16% over the past half a decade (Fig. 4c), with a decrease in the USA of several hundred structures offset by increases elsewhere (Extended Data Fig. 9). By contrast, the number of wind turbines in the ocean has more than doubled since 2017, probably surpassing the number of oil structures by the end of 2020 (Fig. 4c). China leads the development of offshore wind, with a staggering 900% increase in turbines from 2017 to 2021 (averaging around 950 wind turbines per year), well ahead of projections by the International Energy Agency[40]. The UK and Germany lead offshore wind development in Europe, increasing by 49% and 28%, respectively, since 2017.

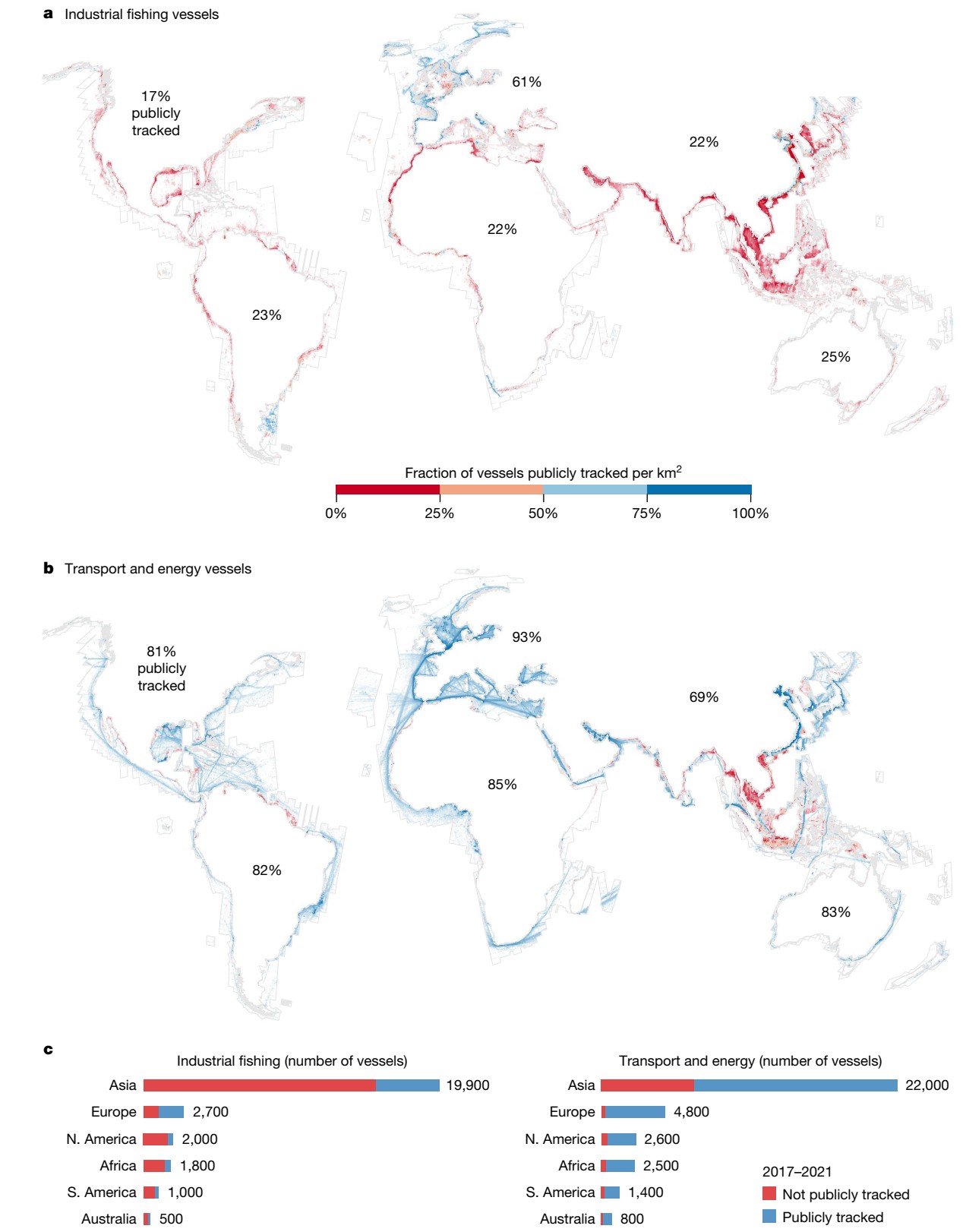

**a** Industrial fishing vessels

17% publicly tracked

61%

22%

22%

23%

25%

Fraction of vessels publicly tracked per km²

0%   25%   50%   75%   100%

**b** Transport and energy vessels

81% publicly tracked

93%

69%

85%

82%

83%

**c**

Industrial fishing (number of vessels)

| | |
|---|---|
| Asia | 19,900 |
| Europe | 2,700 |
| N. America | 2,000 |
| Africa | 1,800 |
| S. America | 1,000 |
| Australia | 500 |

Transport and energy (number of vessels)

| | |
|---|---|
| Asia | 22,000 |
| Europe | 4,800 |
| N. America | 2,600 |
| Africa | 2,500 |
| S. America | 1,400 |
| Australia | 800 |

2017–2021
■ Not publicly tracked
■ Publicly tracked

**Fig. 1 | About 75% of global industrial fishing and 25% of other vessel activity is not publicly tracked. a,b**, Per square kilometre, the average number of industrial fishing vessels (**a**) and shipping, tanker, passenger and support vessels (**b**), from 5 years of satellite SAR imagery. The colour represents the percentage of detected vessels that were matched (blue, publicly tracked) and unmatched (red, not publicly tracked) to known vessel positions from AIS broadcast. **c**, For each continent, the total number of detected vessels and the respective fraction of publicly and not publicly tracked. The outline around the continents (light grey) shows the area of the ocean with available SAR imagery (see Extended Data Fig. 1 for the spatial distribution of images). 'N. America' includes Central American countries. Classification of detected objects was performed with deep learning.

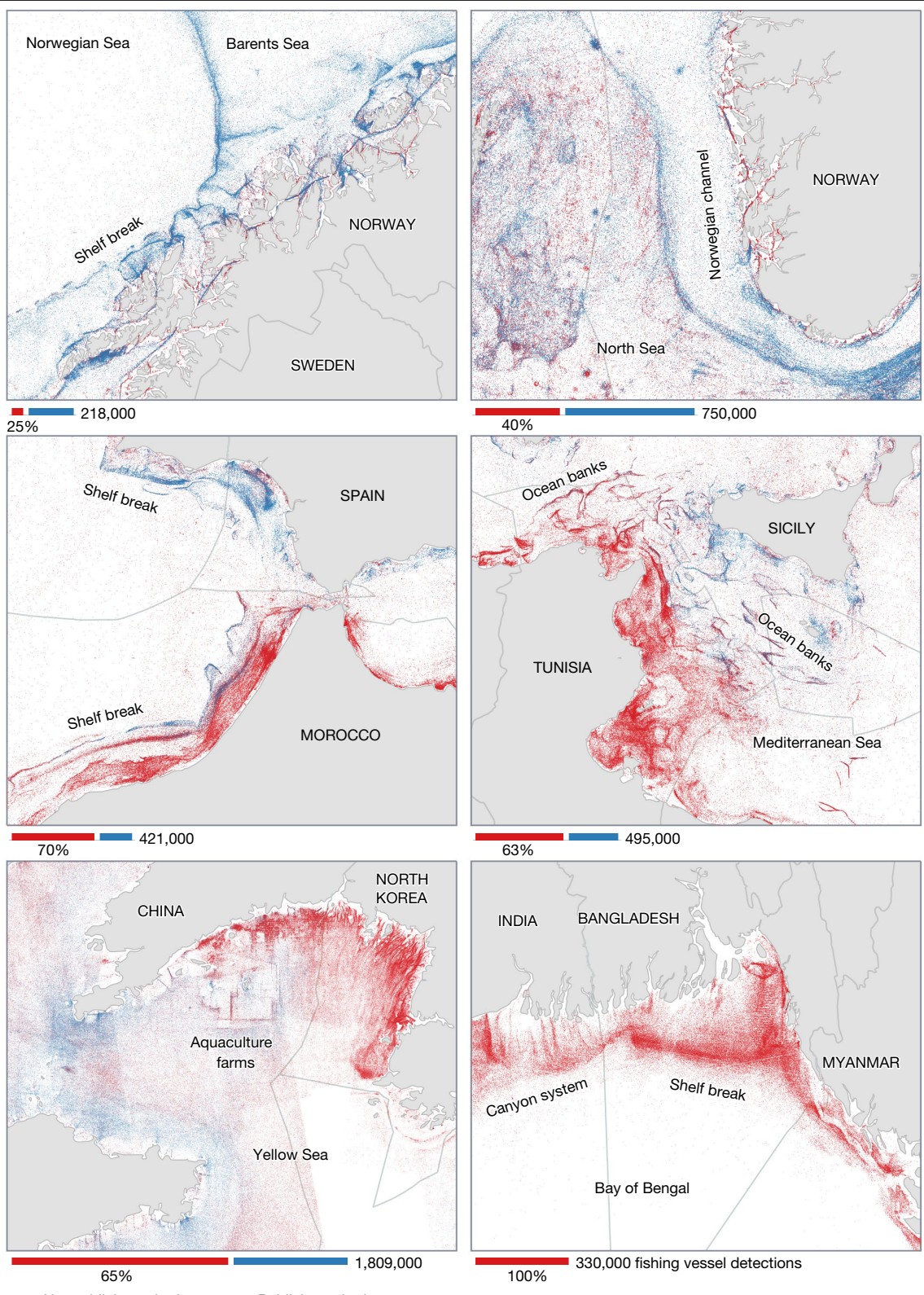

**Fig. 2 | High-resolution mapping reveals detailed patterns of fishing activity not publicly tracked.** Satellite SAR detections of individual vessels during 2017–2021, matched (blue) and unmatched (red) to known vessel positions from AIS broadcast, are classified as fishing or non-fishing vessels with a deep-learning model. Most fishing vessels, usually smaller than 50 m in length, concentrate close to shore and follow bathymetric features, such as the continental shelf break and seabed canyons, or regulatory and political boundaries. Extensive areas of previously unmapped fishing activity are revealed along Northern Africa and South and Southeast Asia. The absolute number of detections in each location depends on the local vessel density and the number of satellite image acquisitions, which varies by region. Depicted numbers may represent a slightly larger area than is shown. This figure shows the level of spatial detail that is possible with our mapping approach. Extended Data Figures 3 and 4 show further examples of high-resolution fishing and non-fishing patterns publicly and not publicly tracked.

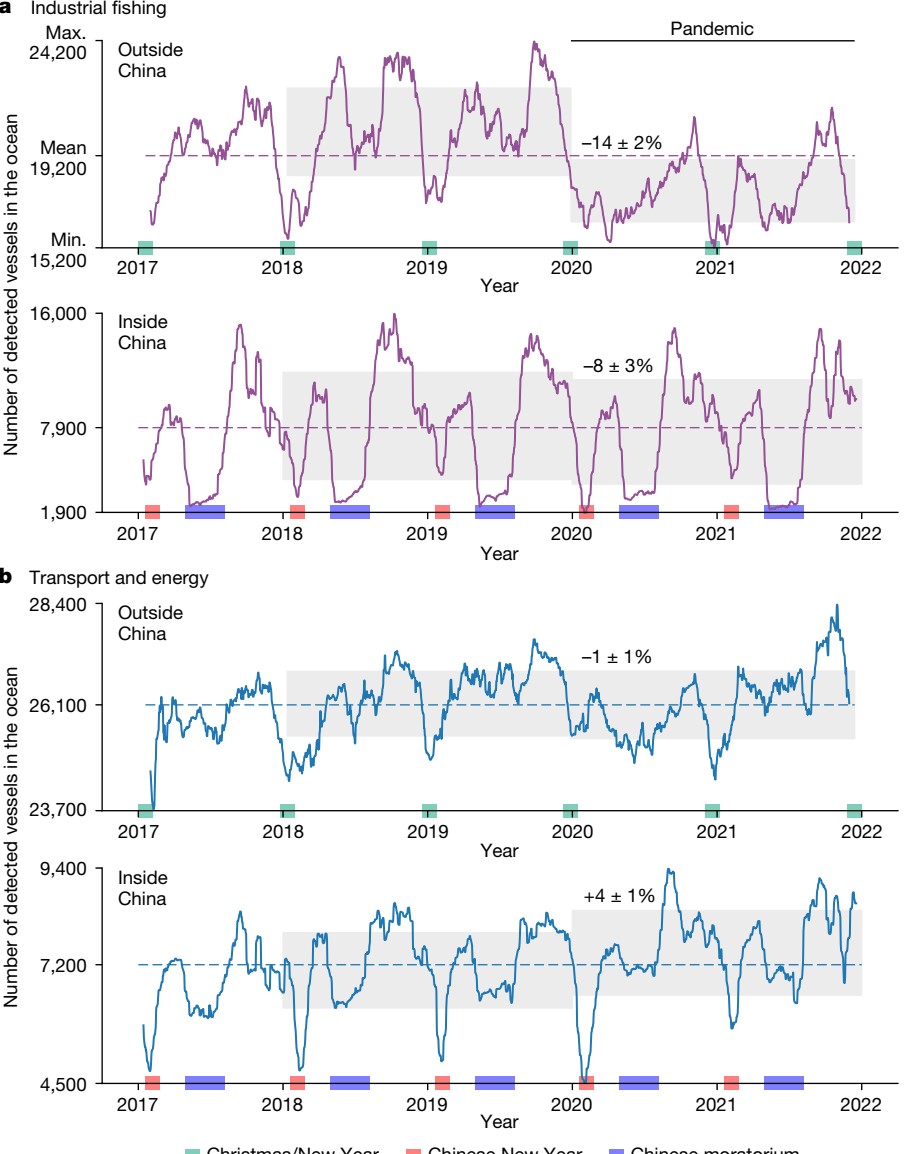

**Fig. 3 | Fishing activity was highly affected by COVID-19, whereas transport and energy continued to grow.** Time series of the average number of vessels over the area covered by SAR from Sentinel-1 (constructed from the average number of detections per satellite overpass at any given location; Methods) showing that COVID-19 greatly affected fishing activity, whereas transport and energy continued to grow. China alone holds nearly 30% of the global fishing fleet and about 21% of transport and energy vessels. **a**, Industrial fishing vessels greater than 15–20 m in length over all EEZs outside China and inside China EEZ. **b**, The same as **a** but for transport-related and energy-related vessels, mostly shipping, tankers, passenger and support. The shaded grey areas indicate the 2-year mean ± 1 s.d., highlighting the effect of the 2020 global pandemic. The numbers show the per cent change with their respective standard error. The combined change (outside + inside) is −12 ± 1% (industrial fishing) and 0 ± 1% (transport and energy); Methods. The coloured boxes highlight the annual cycles in activity related to national holidays and fishing moratoria. The *y* axis shows the maximum, mean and minimum values of detected vessels during 2017–2021.

## Interactions between vessels and fixed infrastructure

A key question for the future is how vessel traffic may be affected by changes in oil and wind infrastructure development. Trawlers, which fish by hauling nets along the seafloor or through the water column and are the most common fishing gear globally, avoid fishing within 1 km of oil structures, most probably to avoid net entanglement (Extended Data Fig. 10a). Other types of fishing, which are at a lower risk of entanglement, are attracted to these structures, probably because they can cause fish to aggregate[41]. Although wind turbines may also aggregate fish, they are less likely to affect industrial fishing in the same way because they are, at present, highly concentrated and, on average, far from shore, where there is less fishing activity (Extended Data Fig. 10b). Also, oil-related vessel traffic has a much wider footprint than wind-related traffic, accounting for five times as much activity globally in 2021 (Fig. 4b and Extended Data Fig. 4).

## Conclusion

Overall, our study reveals the extent of major industrial activities at sea, with fishing being by far the ocean industry with the most activity that is not public. With our freely available dataset and technology, hotspots of potentially illegal activity can now be shown[28] and industrial fishing vessels can be identified that are encroaching on artisanal fishing grounds[17] or other countries' EEZs[27], but at a global scale and accessible to any nation. Maps of global fishing effort can

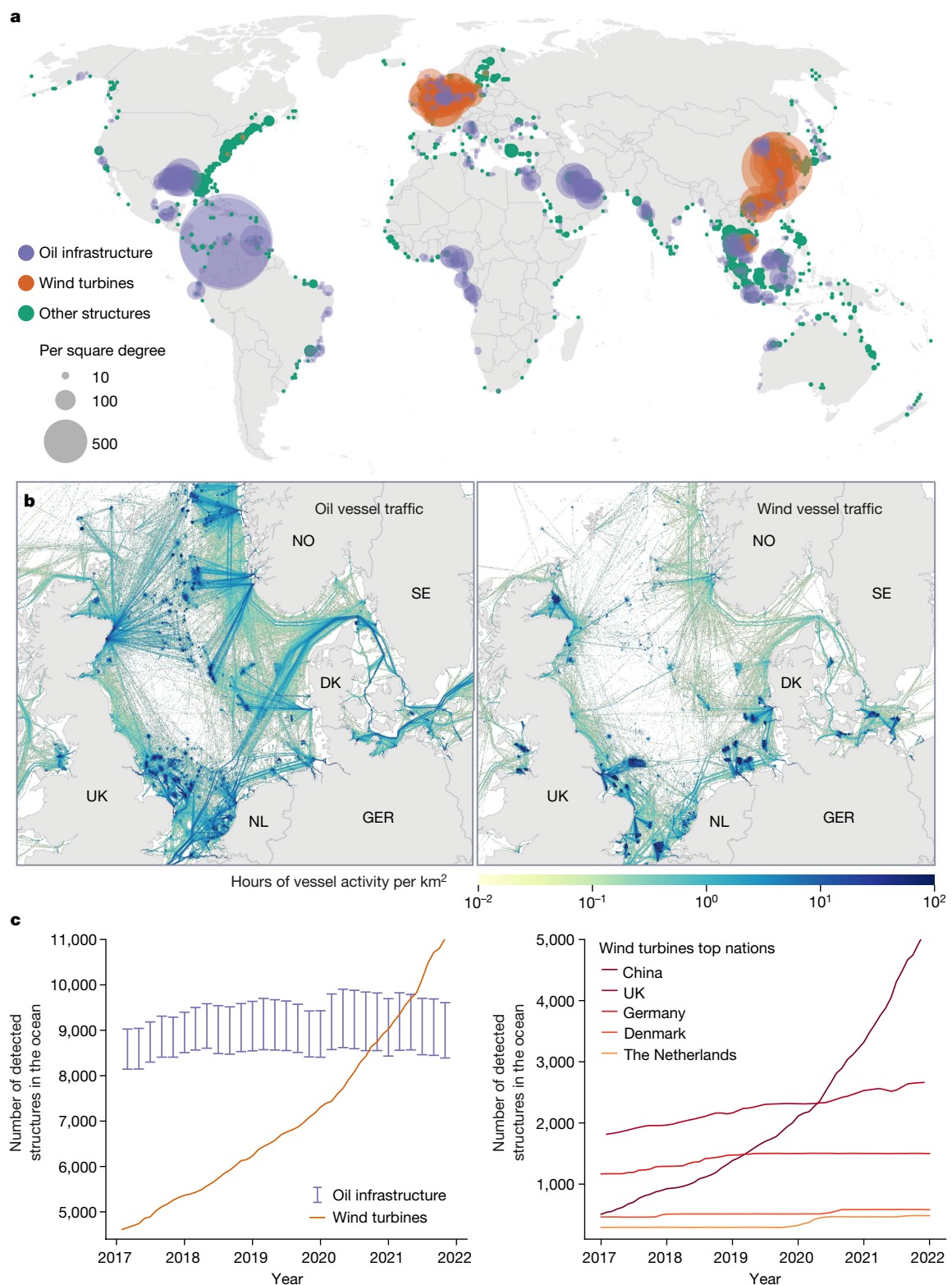

**Fig. 4 | The growing footprint of offshore development extends far beyond the fixed infrastructure. a**, Global map of offshore development, showing oil infrastructure in major oil-producing areas, wind farms and other human-made structures (such as piers, power lines and aquaculture). Circles are proportional to the number of structures per square-degree grid cell at the end of 2021. **b**, One year of vessel traffic associated with offshore infrastructure in the North Sea. These vessels were all broadcasting and interacted with a detected oil or wind structure at some point during 2021 (the vessels were within 200 m of an offshore structure for at least 2 h at a speed of 0 knots). Globally,

in 2021, nearly 4,140,000 h of vessel activity were associated with oil platforms and around 792,500 h with wind turbines. GER, Germany; DK, Denmark; NL, The Netherlands; NO, Norway; SE, Sweden. **c**, Evolution of the number of (fixed) oil and wind structures in the ocean and the leading nations in wind development. Extended Data Figure 9 shows the leading nations in oil development. Error bars define a lower bound considering only high-confidence detections of oil structures and an upper bound including detections with lower confidence (for example, potential oil structures outside oil-producing areas; Methods).

now include all vessels, not just those based on AIS tracking (which misses about three-quarters of large vessels), and with much higher resolution than just EEZs or statistical reporting areas[37,38,42]. Our data can also help to quantify the scale of greenhouse gas emissions from vessel traffic and offshore development, which may help to inform policies on reducing greenhouse gas emissions.

This picture of human activity also presents a snapshot of how industrial use of the ocean is changing. Although COVID-19 may have had a dominant role in depressing fishing activity, fishing still decreased far more than other ocean industries. This slowdown is in line with a long-term decline in the relative importance of fishing in the ocean[43]. Since the 1980s, global marine fish catch has been relatively unchanged as most fisheries are already fished to capacity[11]. As a result, global fishing effort, which has increased several fold since 1950, increased only slightly in recent years[42]. Many countries that have reformed their fisheries show an actual decline in their fishing effort[44]. The decrease highlighted in this study may reflect this longer trend and we may already have seen the peak of fishing activity in the past decade. By contrast, transport and energy vessel traffic may continue to expand, following trends in global trade and the rapid development of renewable energy infrastructure. In this scenario, changes to marine ecosystems brought by infrastructure and vessel traffic may rival fishing in impact[43], and an accurate mapping of these activities is fundamental to understanding and managing future human activities in the ocean.

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

## Methods

### SAR imagery

SAR imaging systems have proved to be the most consistent option for detecting vessels at sea[45,46]. SAR is unaffected by light levels and most weather conditions, including daylight or darkness, clouds or rain. By contrast, some other satellite sensors, such as electro-optical imagery, rely on sunlight and/or the infrared radiation emitted by objects on the ground and can therefore be confounded by cloud cover, haze, weather events and seasonal darkness at high latitudes.

We used SAR imagery from the Copernicus Sentinel-1 mission of the European Space Agency (ESA) (https://sentinel.esa.int/web/sentinel/user-guides/sentinel-1-sar). The images are sourced from two satellites (S1A and, formerly, S1B, which stopped operating in December 2021) that orbit 180° out of phase with each other in a polar, sun-synchronous orbit. Each satellite has a repeat cycle of 12 days, so that—together—they provide a global mapping of coastal waters around the world approximately every 6 days. The number of images per location, however, varies greatly depending on mission priorities, latitude and degree of overlap between adjacent satellite passes (https://sentinels.copernicus.eu/web/sentinel/missions/sentinel-1/observation-scenario). Spatial coverage also varies over time and is improved with the addition of S1B in 2016 and the acquisition of more images in later years (Extended Data Fig. 1). Our data consist of dual-polarization images (VH and VV) from the Interferometric Wide (IW) swath mode, with a resolution of about 20 m. We used the Ground Range Detected (GRD) Level-1 product provided by Google Earth Engine (https://developers.google.com/earth-engine/datasets/catalog/COPERNICUS_S1_GRD), processed for thermal noise removal, radiometric calibration and terrain correction (https://developers.google.com/earth-engine/guides/sentinel1). To eliminate potential noise artefacts[33] that would introduce false detections, we further processed each image by clipping a 500-m buffer off the borders. We selected all SAR scenes over the ocean from October 2016 to February 2022, comprising 753,030 images of 29,400 × 24,400 pixels each on average.

### Visible and NIR imagery

For optical imagery, we used the Copernicus Sentinel-2 (S2) mission of the ESA (https://sentinels.copernicus.eu/web/sentinel/user-guides/sentinel-2-msi). These twin satellites (S2A and S2B) also orbit 180° out of phase and carry a wide-swath, high-resolution, multispectral imaging system, with a combined global 5-day revisit frequency. Thirteen spectral bands are sampled by the S2 Multispectral Instrument (MSI): visible (RGB) and NIR at 10 m, red edge and SWIR at 20 m, and other atmospheric bands at 60-m spatial resolution. We used the RGB and NIR bands from the Level-1C product provided by Google Earth Engine (https://developers.google.com/earth-engine/datasets/catalog/COPERNICUS_S2) and we excluded images with more than 20% cloud coverage using the QA60 bitmask band with cloud mask information. We analysed all scenes that contained a detected offshore infrastructure during our observation period, comprising 2,494,370 images of 10,980 × 10,980 pixels each on average (see the 'Infrastructure classification' section).

### AIS data

AIS data were obtained from satellite providers ORBCOMM and Spire. In total, using Global Fishing Watch's data pipeline[5], we processed 53 billion AIS messages. From those data, we extracted the locations, lengths and identities of all AIS devices that operated near the SAR scenes around the time the images were taken; we did so by interpolating between AIS positions to identify where vessels probably were at the moment of the image, as described in ref. 47. Identities of vessels in the AIS were based on methods in ref. 5 and revised in ref. 26.

### Environmental and physical data

To classify vessels detected with SAR as fishing and non-fishing, we constructed a series of global environmental fields that were used as features in our model. Each of these rasters represents an environmental variable over the ocean at 1-km resolution. Data were obtained from the following sources: chlorophyll data from the NASA Ocean Biology Processing Group (https://oceancolor.gsfc.nasa.gov/data/10.5067/ORBVIEW-2/SEAWIFS/L2/IOP/2018), sea-surface temperature and currents from the Copernicus Global Ocean Analysis and Forecast System (https://doi.org/10.48670/moi-00016), distance to shore from NASA OBPG/PacIOOS (http://www.pacioos.hawaii.edu/metadata/dist2coast_1deg_ocean.html), distance to port from Global Fishing Watch (https://globalfishingwatch.org/data-download/datasets/public-distance-from-port-v1) and bathymetry from GEBCO (https://www.gebco.net/). EEZ boundaries used in our analysis and maps are from Marine Regions[48].

### Vessel detection by SAR

Detecting vessels with SAR is based on the widely used constant false alarm rate (CFAR) algorithm[46,49,50], a standard adaptive threshold algorithm used for anomaly detection in radar imagery. This algorithm is designed to search for pixel values that are unusually bright (the targets) compared with those in the surrounding area (the sea clutter). This method sets a threshold that depends on the statistics of the local background, sampled with a set of sliding windows. Pixel values above the threshold constitute an anomaly and are probably samples from a target. Our modified two-parameter CFAR algorithm evaluates the mean and standard deviation of backscatter values, delimited by a 'ring' composed of an inner window of 200 × 200 pixels and an outer window of 600 × 600 pixels. The best separation between the ocean and the targets is accomplished by the vertical–horizontal (VH) polarization band, which shows relatively low polarized returns over flat areas (ocean surface) compared with volumetric objects (vessels and infrastructure)[45]:

$$x_{px} > \mu_b + \sigma_b n_t \Leftrightarrow \text{anomaly}$$

in which $x_{px}$ is the backscatter value of the centre pixel, $\mu_b$ and $\sigma_b$ are the mean and standard deviation of the background, respectively, and $n_t$ is a time-dependent threshold.

To maximize detection performance, we determined the sizes of the windows empirically, based on the fraction of detected vessels (broadcasting AIS) with length between 15 m and 20 m. A key feature of our two-parameter CFAR algorithm is the ability to specify different thresholds for different times. This adjustment is needed because the statistical properties of the SAR images provided by Sentinel-1 vary with time as well as by satellite (S1A and S1B). We thus found that the ocean pixels for both the mean and the standard deviation of the scenes changed, requiring different calibrations of the CFAR parameters for five different time intervals during which the statistics of the images remained relatively constant: January 2016 to October 2016 ($n_{S1A} = 14$, $n_{S1B} =$ none); September 2016 to January 2017 (14, 18); January 2017 to March 2018 (14, 17); March 2018 to January 2020 (16, 19); and January 2020 to December 2021 (22, 24). The five detection thresholds were calibrated to obtain a consistent detection rate for the smaller vessels across the entire Sentinel-1 archive (60% detection of vessels 15–20 m in length). The relative simplicity of our approach allowed us to reprocess the full archive of Sentinel-1 imagery several times to empirically determine the optimal parameters for detection.

To implement our SAR detection algorithm, we used the Python API of Google Earth Engine (https://developers.google.com/earth-engine/tutorials/community/intro-to-python-api), a planetary-scale platform for analysing petabytes of satellite imagery and geospatial datasets. For processing, analysing and distributing our data products, our detection

workflow uses Google's cloud infrastructure for big data, including Earth Engine, Compute Engine, Cloud Storage and BigQuery.

## Vessel presence and length estimation

To estimate the length of every detected object and also to identify when our CFAR algorithm made false detections, we designed a deep convolutional neural network (ConvNet) based on the modern ResNet (Residual Networks) architecture[51]. This single-input/multi-output ConvNet takes dual-band SAR image tiles of 80 × 80 pixels as input and outputs the probability of object presence (a binary classification task) and the estimated length of the object (a regression task).

To analyse every detection, we extracted a small tile from the original SAR image that contained the detected object at the centre and that preserved both polarization bands (VH and VV). Our inference data therefore consisted of more than 62 million dual-band image tiles to classify. To construct our training and evaluation datasets, we used SAR detections that matched to AIS data with high confidence (see the 'SAR and AIS integration' section), including a variety of challenging scenarios such as icy locations, rocky locations, low-density and high-density vessel areas, offshore infrastructure areas, poor-quality scenes, scenes with edge artefacts and so on (Extended Data Fig. 11). To inspect and annotate these samples, we developed a labelling tool and used domain experts, cross-checking annotations from three independent labellers on the same samples and retaining the high-confidence annotations. Overall, our labelled data contained about 12,000 high-quality samples that we partitioned into the training (80%, for model learning and selection) and test (20%, for model evaluation) sets.

For model learning and selection, we followed a training–validation scheme that uses fivefold cross-validation (https://scikit-learn.org/stable/modules/cross_validation.html), in which, for each fold (a training cycle), 80% of the data is reserved for model learning and 20% for model validation, with the validation subset non-overlapping across folds. Performance metrics are then averaged across folds for model assessment and selection, and the final model evaluation is performed on the holdout test set. Our best model achieved on the test set an F1 score of 0.97 (accuracy = 97.5%) for the classification task and a $R^2$ score of 0.84 (RMSE = 21.9 m, or about 1 image pixel) for the length-estimation task.

## Infrastructure detection

To detect offshore infrastructure, we used the same two-parameter CFAR algorithm developed for vessel detection, with two fundamental modifications. First, to remove non-stationary objects, that is, most vessels, we constructed median composites from SAR images within a 6-month time window. Because stationary objects are repeated across most images, they are retained with the median operation, whereas non-stationary objects are excluded. We repeated this procedure for each month, generating a monthly time series of composite images. The temporal aggregation of images also reduces the background noise (the sea clutter) while enhancing the coherent signals from stationary objects[33]. Second, we empirically adjusted the sizes of the detection window. As some offshore infrastructure is usually arranged in dense clusters, such as wind farms following a grid-like pattern, we reduced the spatial windows to avoid 'contamination' from neighbouring structures. It is also common to find smaller structures such as weather masts placed between some of the wind turbines. We found that an inner window of 140 × 140 pixels and outer window of 200 × 200 pixels was optimal for detecting every object in all wind farms and oil fields that we tested, including Lake Maracaibo, the North Sea and Southeast Asia, areas known for their high density of structures (Extended Data Fig. 7).

## Infrastructure classification

To classify every detected offshore structure, we used deep learning. We designed a ConvNet based on the ConvNeXt architecture[52]. A key difference from the 'vessel presence and length estimation' model, besides using a different architecture, is that this model is a multi-input/single-output ConvNet that takes two different multiband image tiles of 100 × 100 pixels as input, passes them through independent convolutional layers (two branches), concatenates the resulting feature maps and, with a single classification head, outputs the probabilities for the specified classes: wind infrastructure, oil infrastructure, other infrastructure and noise.

A new aspect of our deep-learning classification approach is the combination of SAR imagery from Sentinel-1 with optical imagery from Sentinel-2. From 6-month composites of dual-band SAR (VH and VV) and four-band optical (RGB and NIR) images, we extracted small tiles for every detected fixed structure, with the respective objects at the centre of the tile. Although both the SAR and optical tiles consist of 100 pixels, they come from imagery with different resolutions: the dual-band SAR tile has a spatial resolution of 20 m per pixel and the four-band optical tile is 10 m per pixel. This variable resolution not only provides information with different levels of granularity but also yields different fields of view.

From our inference data for infrastructure classification, which consisted of nearly six million multiband images, we constructed the labelled data by integrating several sources of ground truth for 'oil and gas' and 'offshore wind': from the Bureau of Ocean Energy Management (https://www.data.boem.gov/Main/HtmlPage.aspx?page=platformStructures), the UK Hydrographic Office (https://www.admiralty.co.uk/access-data/marine-data), the California Department of Fish and Wildlife (https://data-cdfw.opendata.arcgis.com/datasets/CDFW::oil-platforms-ospr-ds357/about) and Geoscience Australia (https://services.ga.gov.au/gis/rest/services/Oil_Gas_Infrastructure/MapServer). Using a labelling approach similar to that of the vessel samples, we also inspected a large number of detections to identify samples for 'other structures' and 'noise' (rocks, small islands, sea ice, radar ambiguities and image artefacts). From all areas known to have some offshore infrastructure (Extended Data Fig. 11), our labelled data contained more than 47,000 samples (45% oil, 41% wind, 10% noise and 4% other) that we partitioned into the training (80%) and test (20%) sets, using the same fivefold cross-validation strategy as for vessels.

Because the same fixed objects appear in several images over time, we grouped the candidate structures for the labelled data into 0.1° spatial bins and sampled from different bins for each data partition, so that the subsets for model learning, selection and evaluation did not contain the same (or even nearby) structures at any point. We also note that, in the few cases in which optical tiles were unavailable, for example, because of seasonal darkness close to the poles, the classification was performed with SAR tiles only (optical tiles were blank). Our best model achieved on the test set a class-weighted average F1 score of 0.99 (accuracy = 98.9%) for the multiclass problem.

## Fishing and non-fishing classification

To identify whether a detected vessel was a fishing or non-fishing boat, we also used deep learning. For this classification task, we used the same underlying ConvNeXt architecture as for infrastructure, modified to process the following two inputs: the estimated length of the vessel from SAR (a scalar quantity) and a stack of environmental rasters centred at the location of the vessel (a multiband image). This multi-input-mixed-data/single-output model passes the raster stack (11 bands) through a series of convolutional layers and combines the resulting feature maps with the vessel-length value to perform a binary classification: fishing or non-fishing.

Two key aspects of our neural-net classification approach differ greatly from conventional image-classification tasks.

First, we are classifying the environmental context in which the vessel in question operates. To do so, we constructed 11 gridded fields (rasters) with a resolution of 0.01° (approximately 1 km per pixel at the equator) and with global coverage. At every pixel, each raster contains contextual information on the following variables: (1) vessel density (based on SAR); (2) average vessel length (based on SAR); (3) bathymetry;

(4) distance from port, (5) and (6) hours of non-fishing-vessel presence (from the AIS) for vessels less than 50 m and more than 50 m, respectively; (7) average surface temperature; (8) average current speed; (9) standard deviation of daily temperature; (10) standard deviation of daily current speed; and (11) average chlorophyll. For every detected vessel, we sampled 100 × 100-pixel tiles from these rasters, producing an 11-band image that we then classified with the ConvNet. Each detection is thus provided with context in an area just over 100 × 100 km. We obtained the fishing and non-fishing labels from AIS vessel identities[26].

Second, our predictions are produced with an ensemble of two models with no overlap in spatial coverage. To avoid leakage of spatial information between the training sets of the two models, and also to maximize spatial coverage, we divided the centre of the tiles into a 1° longitude and latitude grid. We then generated two independent labelled datasets, one containing the tiles from the 'even' and the other from the 'odd' latitude and longitude grid cells. This alternating 1° (the size of the tile) strategy ensures no spatial overlap between tiles across the two sets. We trained two independent models, one for 'even' tiles and another for 'odd' tiles, with each model 'seeing' a fraction of the ocean that the other model does not 'see'. The test set that we used to evaluate both models contains tiles from both 'even' and 'odd' grid cells, with a 0.5° buffer around all the test grid cells removed from all the neighbouring cells (used for training) to ensure spatial independence across all data partitions (no leakage). By averaging the predictions from these two models, we covered the full spatial extent of our detections with independent and complementary spatial information.

Our original test set contained 47% fishing and 53% non-fishing samples. We calibrated the model output scores by adjusting the ratio of fishing to non-fishing vessels in the test set to 1:1 (https://scikit-learn.org/stable/modules/calibration.html). We performed a sensitivity analysis to see how our results changed with different proportions of fishing and non-fishing vessels, 2:1 and 1:2. On average, about 30,000 vessels not publicly tracked were detected at any given time. The calibrated scores with two-thirds fishing vessels predicted that 77% of these vessels were fishing, whereas the calibration with only one-third fishing vessels predicted that 63% of them were fishing vessels. Thus, the total percentage (considering all detections) of fishing and non-fishing vessels not publicly tracked amounts to 72–76% and 21–30%, respectively. Analysts at Global Fishing Watch then reviewed these outputs in different regions of the world to verify its accuracy.

Our training data contained about 120,000 tiles (divided into 'odd' and 'even') that we split into 80% for model learning and 20% for model selection. Our test set for model evaluation contained 14,100 tiles from both 'odd' and 'even' grid cells (Extended Data Fig. 11). The inference data contained more than 52 million tiles (11-band images) with respective vessel lengths that we classified with the two models. Our best model ensemble achieved on the test set a F1 score of 0.91 (accuracy = 90.5%) for the classification task.

## False positives and recall

Because there is no ground-truth data on where vessels are not present, estimating the rate of false positives at the global scale of our vessel detection algorithm is challenging. Although some studies report the total number of false positives, we believe that a more meaningful metric is the 'false positive density' (number of false positives per unit area), which takes into account the actual scale of the study. We estimated this metric by analysing 150 million km² of imagery across all five years in regions with very low density of AIS-equipped vessels (less than 10 total hours in 2018 in a grid cell of 0.1°), in regions far from shore (>20 km) and in the waters of countries that have relatively good AIS use and reception. The number of non-broadcasting vessel detections in these regions serves as the upper limit on the density of false positives, which we estimated as 5.4 detections per 10,000 km². If all of these were false positives, it would suggest a false-positive

rate of about 2% in our data. Because many of these are probably real detections, however, the actual false-positive rate is probably lower. Compared with other sources of uncertainties, such as the resolution limitation of the SAR imagery and missing some areas of the ocean (see below), false positives introduce a relatively minor error to our estimations.

To estimate recall (proportion of actual positives correctly identified), we used a method similar to that used in ref. 47. We identified all vessels that had an AIS position very close in time to the image acquisition (<2 min) and should therefore have appeared in the SAR scene; if they were detected in the SAR image, we could match them to the respective AIS-equipped vessels and then identify the AIS-equipped vessels not detected. The recall curve suggests that we are able to detect more than 95% of all vessels greater than 50 m in length and around 80% of all vessels between 25 m and 50 m in length, with the detection rate decaying steeply for vessels smaller than 25 m (Extended Data Fig. 2). However, because our vessel detection relies on a CFAR algorithm with a 600-m-wide window, when vessels are close to one another (<1 km), the detection rate is lower. See the 'Limitations of our study' section for factors influencing detectability.

## SAR and AIS integration

Matching SAR detections to the GPS coordinates of vessels (from AIS records) is challenging because the timestamp of the SAR images and AIS records do not coincide, and a single AIS message can potentially match to several vessels appearing in the image, and vice versa. To determine the likelihood that a vessel broadcasting AIS signals corresponded to a specific SAR detection, we followed the matching approach outlined in ref. 47, with a few improvements. This method draws on probability rasters of where a vessel probably is minutes before and after an AIS position was recorded. These rasters were developed from one year of global AIS data, including roughly 10 billion vessel positions, and computed for six different vessel classes, considering six different speeds and 36 time intervals, leading to 1,296 rasters. This probability raster approach could be seen as a utilization distribution[53]—for each vessel class, speed and time interval—in which the space is relative to the position of the individual.

As described in ref. 47, we combined the before and after probability rasters to obtain the probability distribution of the probable location of each vessel. We then calculated the value of this probability distribution at each SAR detection that a given vessel could match to. This value was then adjusted to account for: (1) the likelihood a vessel was detected and (2) a factor to account for whether the length of the vessel (from Global Fishing Watch's AIS database) is in agreement with the length estimated from the SAR image. The resulting value provides a score for each potential AIS to SAR match, calculated as

$$\text{score} = p L_{\text{detect}} L_{\text{match}}$$

in which $p$ is the value of the probability distribution at the location of the detection (following ref. 47), $L_{\text{match}}$ is a factor that adjusts this score based on length and $L_{\text{detect}}$ is the likelihood of detecting the vessel, defined as

$$L_{\text{detect}} = R(\text{length, spacing}) L_{\text{inside}}$$

in which $R$ is the recall as a function of vessel size and distance to the nearest vessel with an AIS device (Extended Data Fig. 2) and $L_{\text{inside}}$ is the probability that the vessel was in the scene at the moment of the image, obtained by calculating the fraction of a vessel's probability distribution that is within the given SAR scene[47]. Drawing on 2.8 million detections of high-confidence matches (AIS to SAR matches that were unlikely to match to other detections and for which the AIS-equipped vessel had a position within 2 min of the image), we developed a lookup table with the fractional difference between AIS known length and SAR

estimated length, discretized in 0.1 difference intervals. Multiplying by this value ($L_{match}$) makes it very unlikely for a small vessel to match to a large detection, or vice versa.

A matrix of scores of potential matches between SAR and AIS is then computed and matches are assigned (by selecting the best option available at the moment) and removed in an iterative procedure, with our method performing substantially better than conventional approaches, such as interpolation based on speed and course[47]. A key challenge for us is deciding on the best score threshold to accept or reject a match, because a threshold that is too low or too high would increase or decrease the likelihood that a given SAR detection is a vessel not publicly tracked. To determine the optimal score, we estimated the total number of vessels with AIS devices that should have appeared in the scenes globally by summing $R$(length, spacing)$L_{inside}$ for all scenes. This value suggests that, globally, 17 million vessels with AIS devices should have been detected in the SAR images. As such, we selected the threshold that provided 17 million matches from the actual detections, that is, $7.4 \times 10^{-6}$.

We refer to ref. 47 for the full description of the raster-based matching algorithm, and the matching code can be found at https://github.com/GlobalFishingWatch/paper-longline-ais-sar-matching.

## Data filtering

Delineating shorelines is difficult because current global datasets do not capture the complexities of all shorelines around the world[54,55]. Furthermore, the shoreline is a dynamic feature that constantly changes with time. To avoid false detections introduced by inaccurately defined shorelines, we filtered out a 1-km buffer from a global shoreline that we compiled using several sources (https://www.ngdc.noaa.gov/mgg/shorelines, https://www.naturalearthdata.com/downloads/10m-physical-vectors/10m-minor-islands, https://data.unep-wcmc.org/datasets/1, https://doi.org/10.1080/1755876X.2018.1529714, https://osmdata.openstreetmap.de/data/land-polygons.html, https://www.arcgis.com/home/item.html?id=ac80670eb213440ea5899bbf92a04998). We used this synthetic shoreline to determine the valid area for detection within each SAR image.

We filtered out areas with a notable concentration of sea ice, which could introduce false detections because ice is a strong radar reflector, often showing up in SAR images with a similar signature to that of vessels and infrastructure. We used a time-variable sea-ice-extent mask from the Multisensor Analyzed Sea Ice Extent – Northern Hemisphere (MASIE-NH), Version 1 (https://nsidc.org/data/g02186/versions/1#qt-data_set_tabs), supplemented with predefined bounding boxes over lower-latitude areas known to have substantial seasonal sea ice, such as the Hudson Bay in Canada, the Sea of Okhotsk north of Japan, the Arctic Ocean, the Bering Sea, selected areas near Greenland, the northern Baltic Sea and South Georgia Islands. No imagery in the mode we processed was available for Antarctic waters.

We also removed repeated objects across several images (that is, fixed structures) from the vessel-detection dataset so as to exclude them from all calculations about vessel activity. This process also removed vessels anchored for a long period of time, so our dataset is more representative of moving vessels than stationary ones.

Another potential source of noise is reflections from moving vehicles on bridges or roads close to shore. Although bridges can be removed from the data through fixed infrastructure analysis, a vehicle moving perpendicular to the satellite path will appear offset. Vehicles visible in SAR can appear more than a kilometre away from the road when moving faster than 100 km per hour on a highway, sometimes appearing in the water. For matching AIS to SAR, we account for this movement in the matching code[47]. Drawing on the global gROADSv1 dataset of roads, we identified every highway and primary road within 3 km of the ocean (including bridges) and then calculated for each image where vehicles would appear if they were travelling 135 km per hour on a highway or 100 km per hour on a primary road. These offsetting positions were turned into polygons that excluded detections within this distance, which eliminated about 1% of detections globally.

A minor source of false positives is 'radar ambiguities' or 'ghosts', which are an aliasing effect caused by the periodic sampling (radar echoes) of the target to form an image. For Sentinel-1, these ghosts are most commonly caused by bright objects and appear offset a few kilometres in the azimuth direction (parallel to the satellite ground track) from the source object. These ambiguities appear separated from their source by an azimuth angle[56] $\psi = \lambda/(2V)$PRF, in which $\lambda$ is the SAR wavelength, $V$ is the satellite velocity and PRF is the SAR pulse repetition frequency, which—in the case of Sentinel-1—ranges from 1 to 3 kHz and is constant across each sub-swath of the image[35]. Thus, we expect the offsets to also be constant across each sub-swath.

To locate potential ambiguities, we calculated the off-nadir angle[35] $\theta_i$ for every detection $i$ and then identified all detections $j$ within 200 m of the azimuth line through each detection as candidate ambiguities. We then calculated the difference in azimuth angles $\psi_{ij}$ for these candidates. To find which of these detentions were potential ambiguities, we binned the calculated off-nadir angles ($\theta_i$) in intervals of 0.1° (approximately 200 m) and built a histogram for each interval by counting the number of detections at different azimuthal offset angles $\psi$, binning $\psi$ at 0.001°. For each interval $\theta_i$, we identified the angle $\psi$ for which there was the maximum number of detections, limiting ourselves to cases in which the number of detections was at least two standard deviations above the background level. As expected, ambiguities appeared at a consistent $\psi$ within each of the three sub-swaths of the IW mode images. For $\theta < 32.41°$, ambiguities occurred at $\psi = 0.363° \pm 0.004°$. For $32.41° < \theta < 36.87°$, ambiguities occurred at $\psi = 0.308° \pm 0.004°$. And for $\theta > 36.85°$, ambiguities occurred at $\psi = 0.359° \pm 0.004°$.

We then flagged all pairs of detections that lay along a line parallel to the satellite ground track and had an angle $\psi$ within the expected values for their respective sub-swath. The smaller (dimmer) object in the pair was then selected as a potential ambiguity. We identified about 120,000 outliers out of 23.1 million detections (0.5%), which we excluded from our analysis.

Ambiguities can also arise from objects on shore. Because, generally, only objects larger than 100 m produce ambiguities in our data, and few objects larger than 100 m on shore regularly move, these ambiguities probably show up in the same location in images at different times. All stationary objects were removed from our analysis of vessels. The analysis of infrastructure also removed these false detections because, in addition to SAR, it draws on Sentinel-2 optical imagery, which is free from these ambiguities.

We defined spatial polygons for the major offshore oil-producing areas and wind-farm regions (Fig. 4a) and we prescribed a higher confidence to the classification of oil and wind infrastructure falling inside these areas and a lower confidence elsewhere. Overall, we identified 14 oil polygons (Alaska, California, Gulf of Mexico, South America, West Africa, Mediterranean Sea, Persian Gulf, Europe, Russia, India, Southeast Asia, East Asia, Australia, Lake Maracaibo) and two wind polygons (Northern Europe, South and East China seas). We defined these polygons through a combination of: (1) global oil regions datasets (https://doi.org/10.18141/1502839, https://www.prio.org/publications/3685); (2) AIS-equipped vessel activity around infrastructure; and (3) visual inspection of satellite imagery. We then used a DBSCAN[57] clustering approach to identify detections over time (within a 50-m radius) that were probably the same structure but their coordinates differed slightly and assigned them the most common predicted label of the cluster. We also filled in gaps for fixed structures that were missing in one time step but detected in the previous and following time steps and dropped detections appearing in a single time step.

## Vessel activity estimation

To convert individual detections of vessel instances to average vessel activity, we first calculated the total number of detections per pixel on

a spatial grid of 1/200° resolution (about 550 m) and then normalized each pixel by the number of satellite overpasses (number of SAR acquisitions per location). To construct a daily time series of average activity, we performed this procedure with a rolling window of 24 days (two times the repeat cycle of Sentinel-1), aggregating the detections over the window and assigning the value to the centre date. We restricted the temporal analysis to only those pixels that had at least 70 of the 24-day periods (out of 77 possible), which included 95% of the total vessel activity in our study area. For individual pixels with no overpass for 24 days, we linearly interpolated the respective time series at the pixel location. Overall, only 0.7% of the activity in our time series is from interpolated values. This approach provides the average number of vessels present in each location at any given time regardless of spatial differences in frequency and number of SAR acquisitions.

### Temporal change estimation

We computed the global and EEZ mean time series of daily average number of vessels and monthly median number of infrastructure. We aggregated the gridded and normalized data over the area sampled by Sentinel-1 during 2017–2021, when the spatial coverage of Sentinel-1 was fairly consistent (Extended Data Fig. 1). From these times series, we then computed yearly means with respective standard deviations. Although absolute values may be sensitive to the spatial coverage, such as buffering out 1 km from shore, the trends and relative changes are robust as (a) they are calculated over a fixed area over the observation period and (b) this area contains well over three-quarters of all industrial activity at sea (corroborated by AIS). We estimated the per cent change in vessel activity owing to the pandemic (difference between means; Fig. 3) and respective standard error by bootstrapping[58] the residuals with respect to the average seasonal cycle, obtaining for industrial fishing: −14 ± 2% (outside China), −8 ± 3% (inside China), −12 ± 1% (globally); and for transport and energy: −1 ± 1% (outside China), +4 ± 1% (inside China), 0 ± 1% (globally). We note that, for visualization purposes, we smoothed the time series of vessels and offshore infrastructure with a rolling median.

### Limitations of our study

Sentinel-1 does not sample most of the open ocean. As our study shows, however, most of the industrial activity is close to shore. Also, farther from shore, more fishing vessels use AIS (60–90%)[59], far more than the average for all fishing vessels (about 25%). Thus, for most of the world, our analysis complemented with AIS data will capture most of the human activity in the global ocean.

We do not classify objects within 1 km of shore, because of ambiguous coastlines and rocks. Nor do we classify objects in much of the Arctic and Antarctic, in which sea ice can create too many false positives; in both regions, however, vessel traffic is either very low (Antarctic) or in countries that have a high adoption of the AIS (northern European or northern North American countries). The bulk of industrial activities occurs several kilometres from shore, such as fishing along the continental shelf break, ocean transport over shipping lanes and offshore development in medium-to-large oil rigs and wind farms. Also, much of the vessel activity within 1 km of shore is by smaller boats, such as pleasure crafts.

Vessel detection by SAR imagery is limited primarily by the resolution of the images (about 20 m in the case of the Sentinel-1 IW GRD product). As a result, we miss most vessels less than 15 m in length, although an object smaller than a pixel can still be seen if it is a strong reflector, such as a vessel made of metal rather than wood or fibreglass. Especially for smaller vessels (<25 m), detection also depends on wind speed and the state of the ocean[60], as a rougher sea surface will produce higher backscatter, making it difficult to separate a small target from the sea clutter. Conversely, the higher the radar incidence angle, the higher the probability of detection[60], as less backscatter from the background will be received by the antenna. The vessel orientation relative to the

satellite antenna also matters, as a vessel perpendicular to the radar line of sight will have a larger backscatter cross-section, increasing the probability of being detected.

Our estimates of vessel length are limited by the quality of the ground-truth data. Although we selected only high-confidence AIS to SAR matches to construct our training data, we found that some AIS records contained an incorrectly reported length. These errors, however, resulted in only a small fraction of imprecise training labels, and deep-learning models can accommodate some noise in the training data[61].

Our fishing classification may be less accurate in certain regions. In areas of high traffic from pleasure crafts and other service boats, such as near cities in wealthy countries and in the fjords of Norway and Iceland, some of these smaller craft might be misclassified as fishing vessels. Conversely, some misclassification of fishing vessels as non-fishing vessels is expected in areas in which all activity is not publicly tracked, such as Southeast Asia. More importantly, however, is that many industrial fishing vessels are between 10 and 20 m in length, and the recall of our model falls off quickly within these lengths. As a result, the total number of industrial fishing vessels is probably substantially higher than what we detect. Because our model uses vessel length from SAR, it may be possible to use methods similar to those in ref. 47 to estimate the number of missing vessels. Future work can address this challenge.

Overall, our study probably underestimates the concentration of fishing in Asian waters and Chinese fisheries, in which we see areas of vessel activity being 'cut off' by the edge of the Sentinel-1 footprint. And because we miss very small vessels (for example, most artisanal fishing) that are less likely to carry AIS devices, the global estimate of activity not publicly tracked presented here is probably higher. Algorithmic improvements can capture the first kilometre from shore and the inclusion of more SAR satellites in the coming years (two more ESA Sentinel-1 satellites and NASA's NISAR mission) will allow us to apply this method more broadly to build on this map and capture all activity at sea.

## Data availability

All vessel and infrastructure data are freely available through the Global Fishing Watch data portal at https://globalfishingwatch.org/datasets-and-code. All data to reproduce this study can be downloaded from https://doi.org/10.6084/m9.figshare.24309475 (statistical analysis and figures) and https://doi.org/10.6084/m9.figshare.24309469 (model training and evaluation).

## Code availability

All code developed in this study for SAR detection, deep-learning models and analysis is open source and freely available at https://github.com/GlobalFishingWatch/paper-industrial-activity.

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

**Acknowledgements** This work was funded by Bloomberg Philanthropies, National Geographic Pristine Seas and Oceankind. We thank D. Kroodsma for reviewing the manuscript. We thank the European Space Agency (ESA) for making the radar and optical imagery freely available. Google provided in kind computing resources and technical support. All maps were generated using Python (https://www.python.org) with the open-source visualization libraries PySeas (https://github.com/GlobalFishingWatch/pyseas), Matplotlib (https://matplotlib.org) and Cartopy (https://scitools.org.uk/cartopy).

**Author contributions** F.S.P. led the writing, with input from D.K. and J.R. and suggestions from all authors. D.K., F.S.P., P.H. and C.T. conceived the study. D.K. oversaw the project and secured most of the funding. F.S.P. built the detector, with contributions from T.H., C.T., D.K. and P.H. F.S.P. and T.H. built the deep-learning models. F.S.P. and D.K. performed the main analyses, supported by P.D. F.S.P., P.D., C.T. and J.C. reviewed the offshore infrastructure. P.D., S.O., L.M. and D.K. reviewed the vessel detections and fishing classifications. F.S.P., D.K., P.D. and L.M. performed the data labelling. F.S.P. made most of the figures, with D.K. and P.D. contributing further figures. T.H. and D.K. developed the SAR to AIS matching. All authors discussed the results.

**Competing interests** The authors declare no competing interests.

**Additional information**
**Correspondence and requests for materials** should be addressed to Fernando S. Paolo.

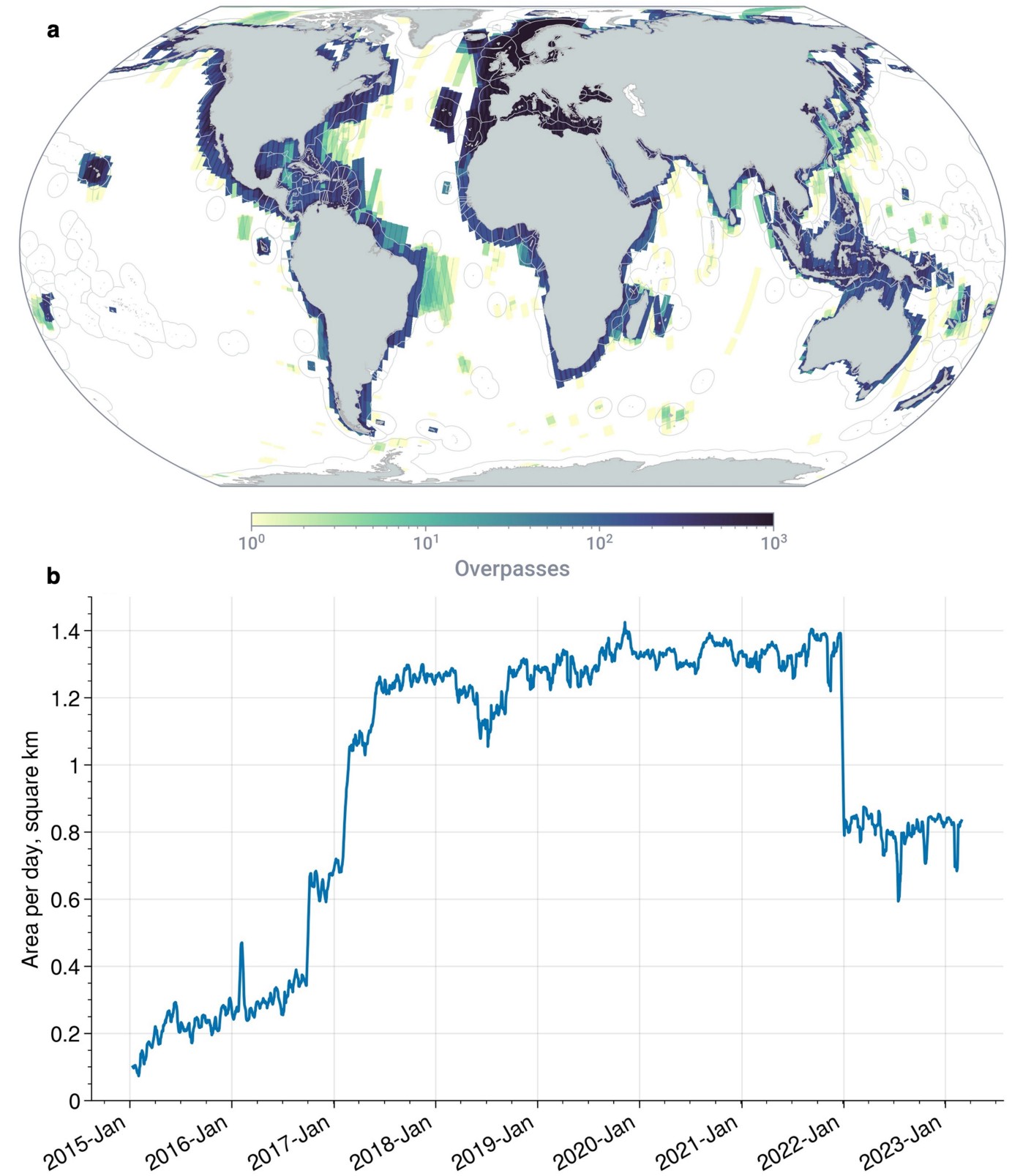

**Extended Data Fig. 1 | The Sentinel-1 SAR imagery (IW GRD product) covers most coastal waters but does not sample most of the open ocean. a**, The extent and frequency of SAR acquisitions is determined by the mission priorities. **b**, The area of the ocean imaged every day by the Sentinel-1 GRD product (using a 12-day rolling average) depended on whether one satellite was imaging the ocean (S1A, October 2014 to present) or two (S1A and S1B, September 2016 to December 2017). S1B stopped operating on 23 December 2021.

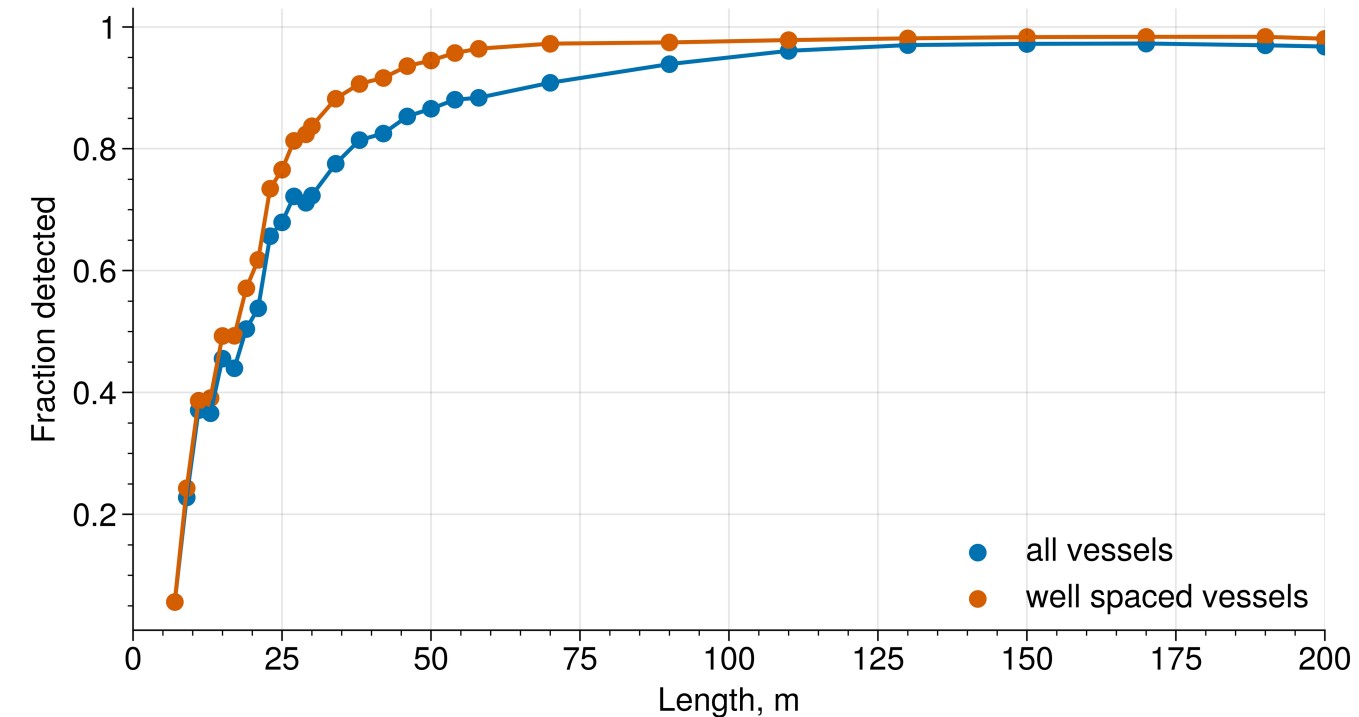

**Extended Data Fig. 2 | The Sentinel-1 detection model is able to detect most industrial vessels.** The recall curve (fraction of actual positives correctly detected) for our Sentinel-1 detection model as a function of vessel length shows that vessels spaced far apart (>1 km distance, constituting 79% of all vessel detections) have higher recall than all vessels combined. For vessels smaller than 25 m, detection performance decays steeply with vessel size.

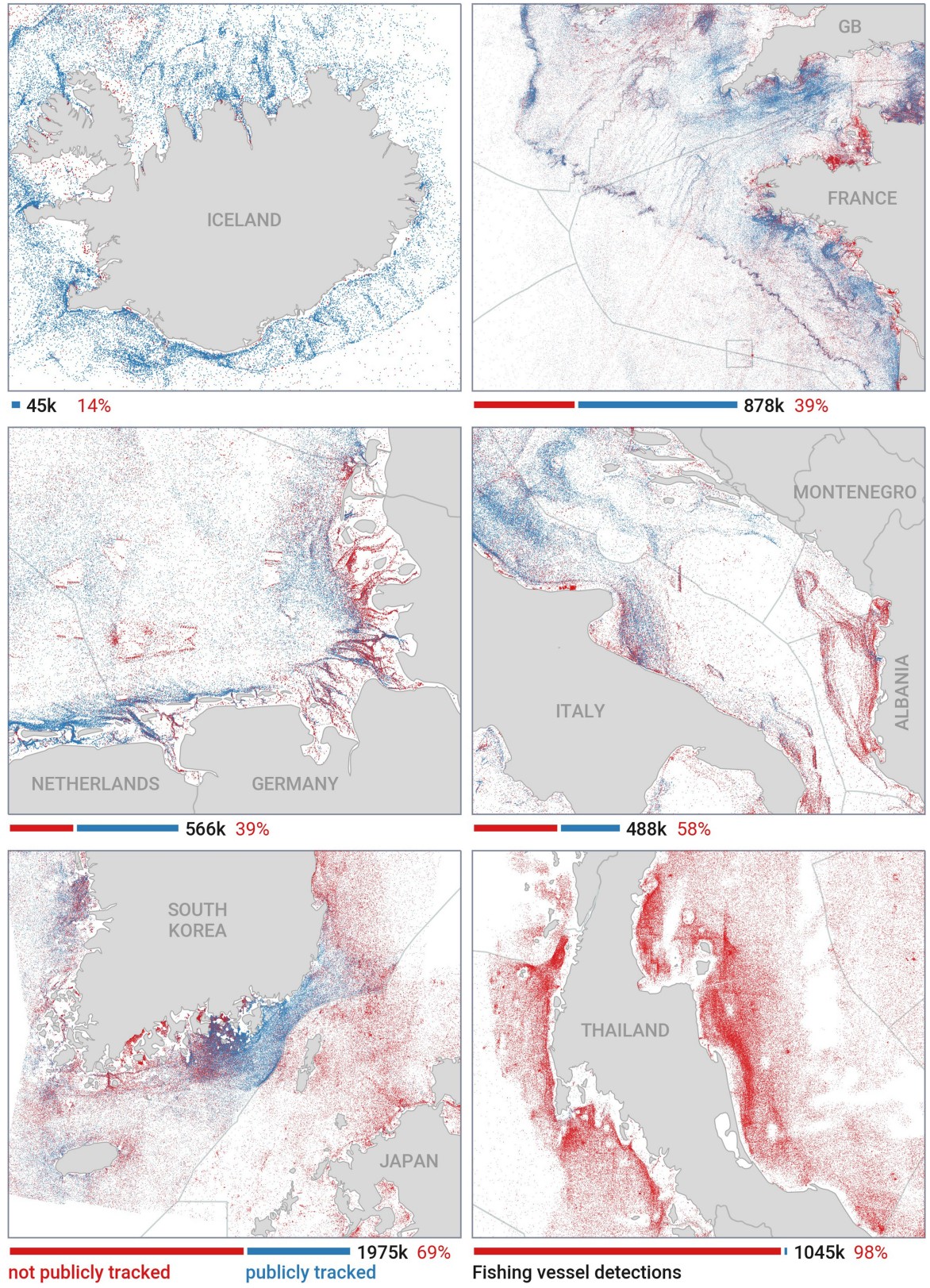

■ 45k 14%

▬▬▬ ▬▬▬▬▬▬ 878k 39%

▬▬▬ ▬▬▬▬▬▬ 566k 39%

▬▬▬ ▬▬▬▬ 488k 58%

▬▬▬▬▬▬▬▬ ▬▬▬ 1975k 69%

▬▬▬▬▬▬▬▬▬▬▬▬▬▬ ■ 1045k 98%

**not publicly tracked**   **publicly tracked**      Fishing vessel detections

**Extended Data Fig. 3 | Fishing vessel activity at sea is shown with an unprecedented level of detail by satellite mapping and deep learning.** Fishing vessels tend to aggregate along bathymetric features. Each dot represents a detected vessel during 2017–2021. The colours represent detections matched (blue, publicly tracked) and unmatched (red, not publicly tracked) to known vessel positions from the AIS. The number of detections in each location depends on the local density of vessels, as well as the number of SAR acquisitions.

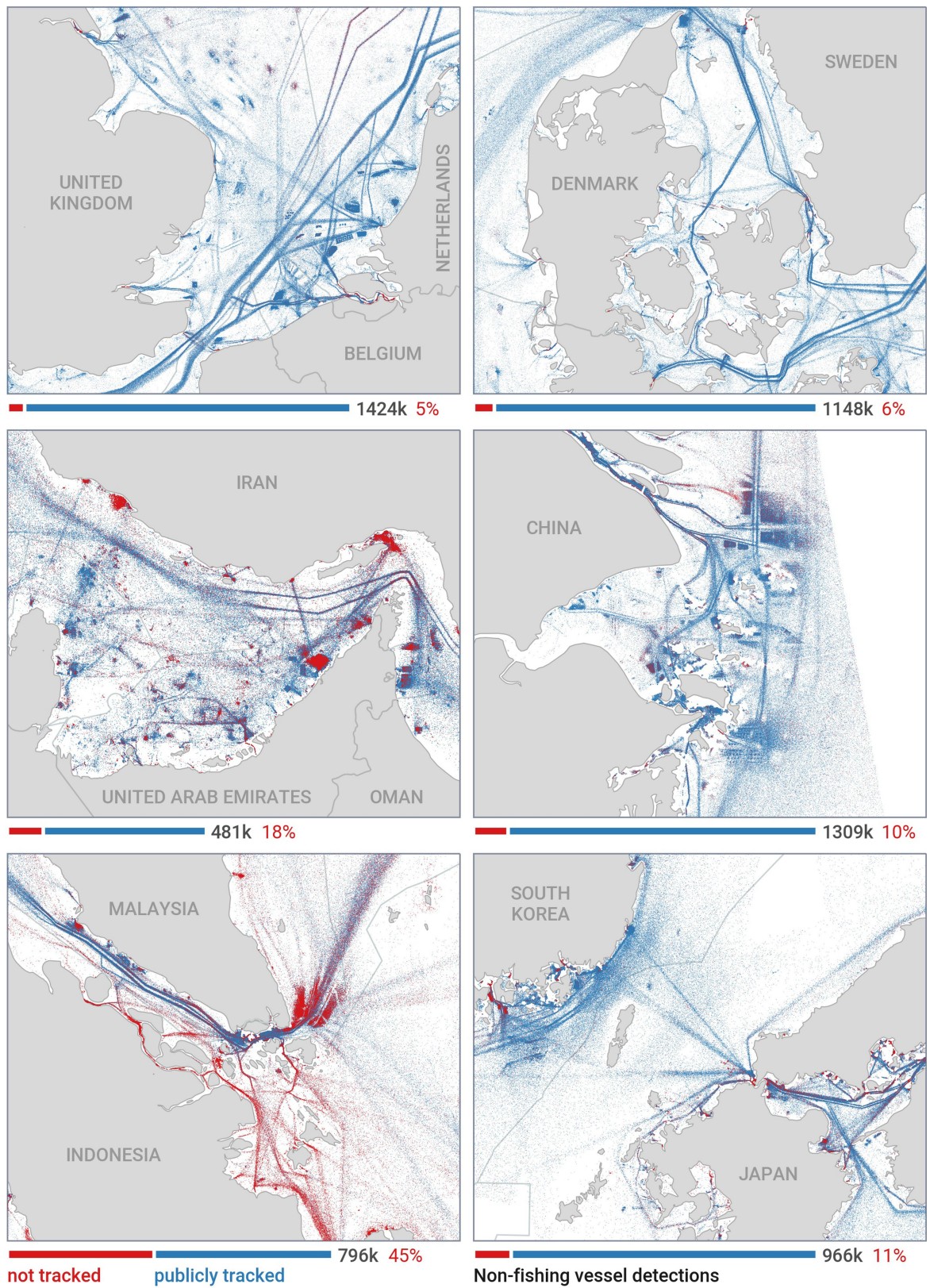

**UNITED KINGDOM** · **NETHERLANDS** · **BELGIUM**
1424k 5%

**SWEDEN** · **DENMARK**
1148k 6%

**IRAN** · **UNITED ARAB EMIRATES** · **OMAN**
481k 18%

**CHINA**
1309k 10%

**MALAYSIA** · **INDONESIA**
796k 45%

not tracked  publicly tracked

**SOUTH KOREA** · **JAPAN**
966k 11%

Non-fishing vessel detections

**Extended Data Fig. 4 | Transport and energy vessel activity at sea is shown with an unprecedented level of detail by satellite mapping and deep learning.** Transport and energy vessels usually follow major routes (for example, shipping lanes). Each dot represents a detected vessel during 2017–2021. The colours represent detections matched (blue, publicly tracked) and unmatched (red, not publicly tracked) to known vessel positions from the AIS. The number of detections in each location depends on the local density of vessels, as well as the number of SAR acquisitions.

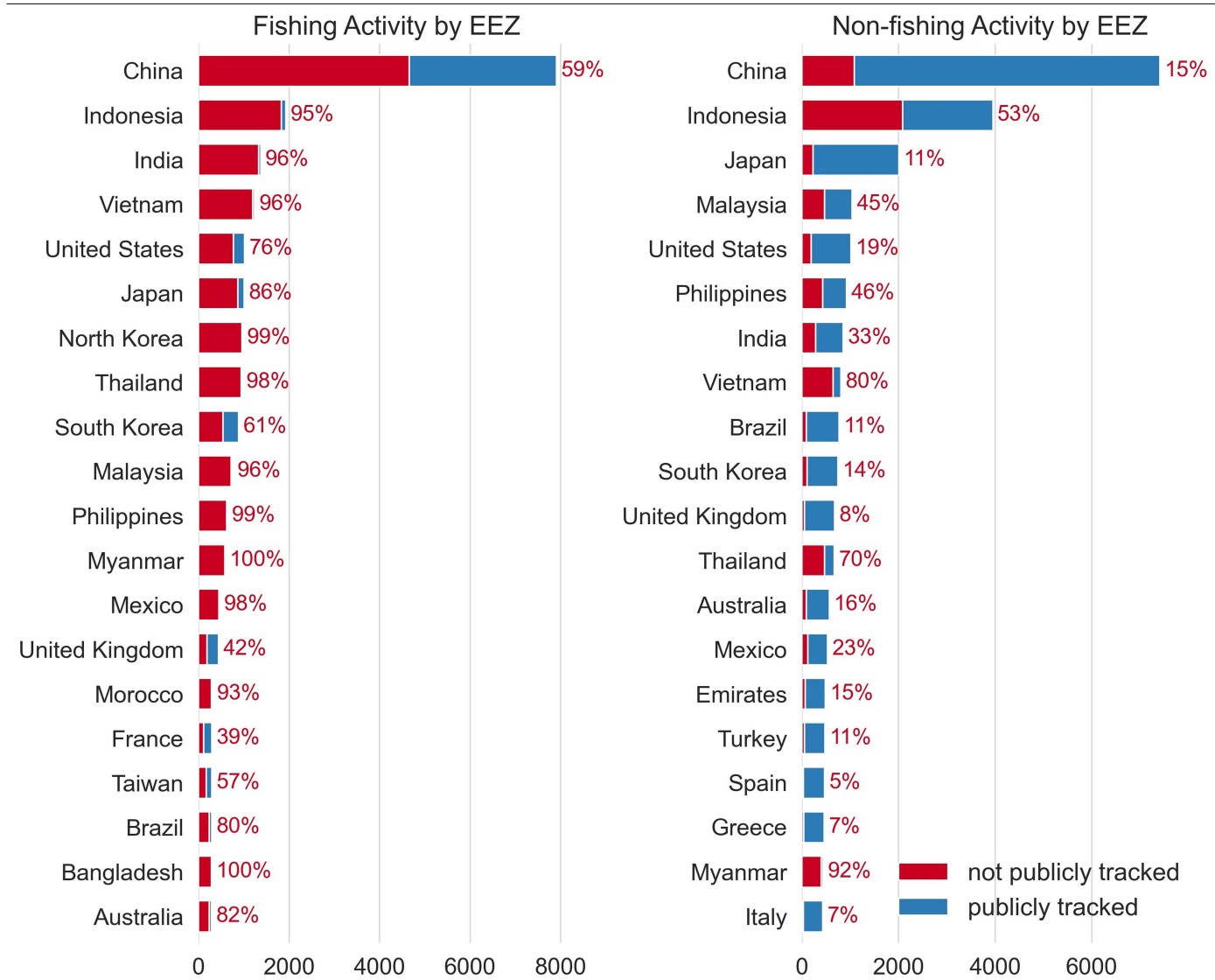

**Extended Data Fig. 5 | Leading nations with most fishing and non-fishing vessel activity.** The bars represent the average number of detections per satellite overpass at any location in the EEZ during 2017–2021. Percentages are the fraction of detections unmatched to known vessel locations from the AIS (activity missing from public monitoring systems).

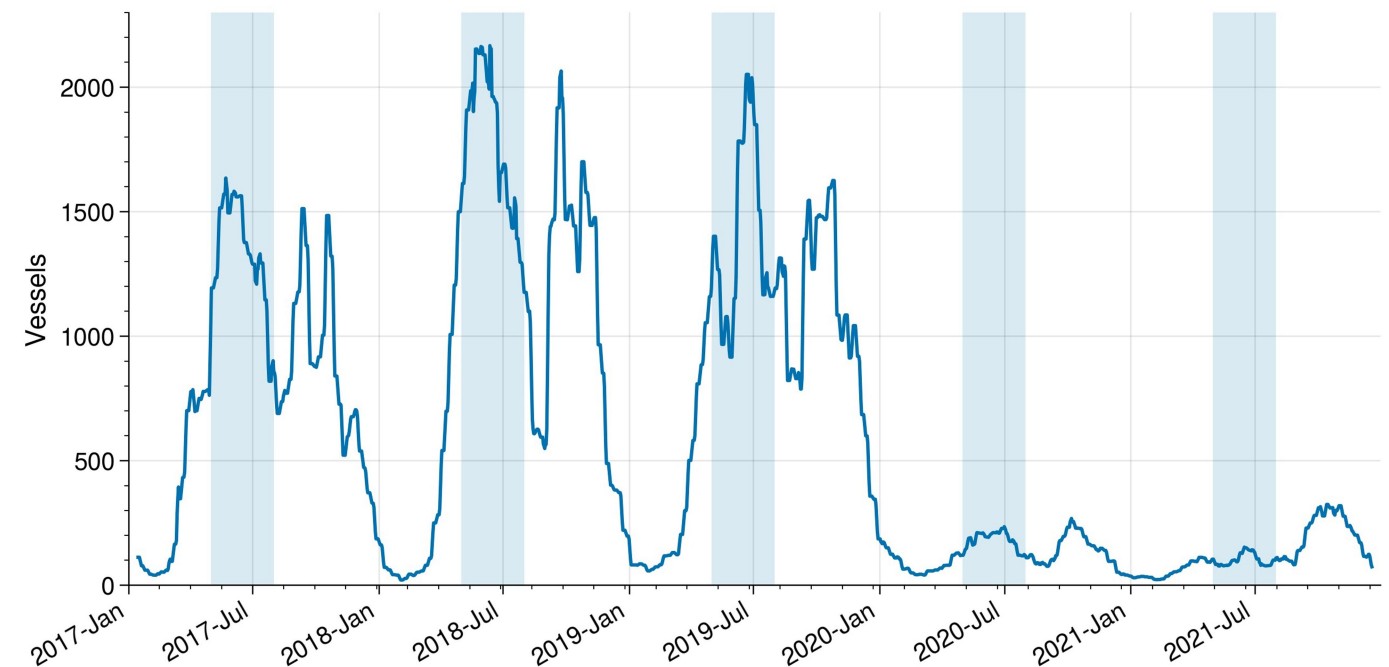

**Extended Data Fig. 6 | In the western North Korean EEZ, peaks of fishing vessel activity coincide with Chinese moratoria on industrial fishing.** Fishing activity in western North Korea waters increases coinciding with the Chinese fishing moratoria (vertical stripes). There is a substantial decrease in overall vessel activity during the COVID-19 pandemic (2020–2021), when North Korea shut its borders.

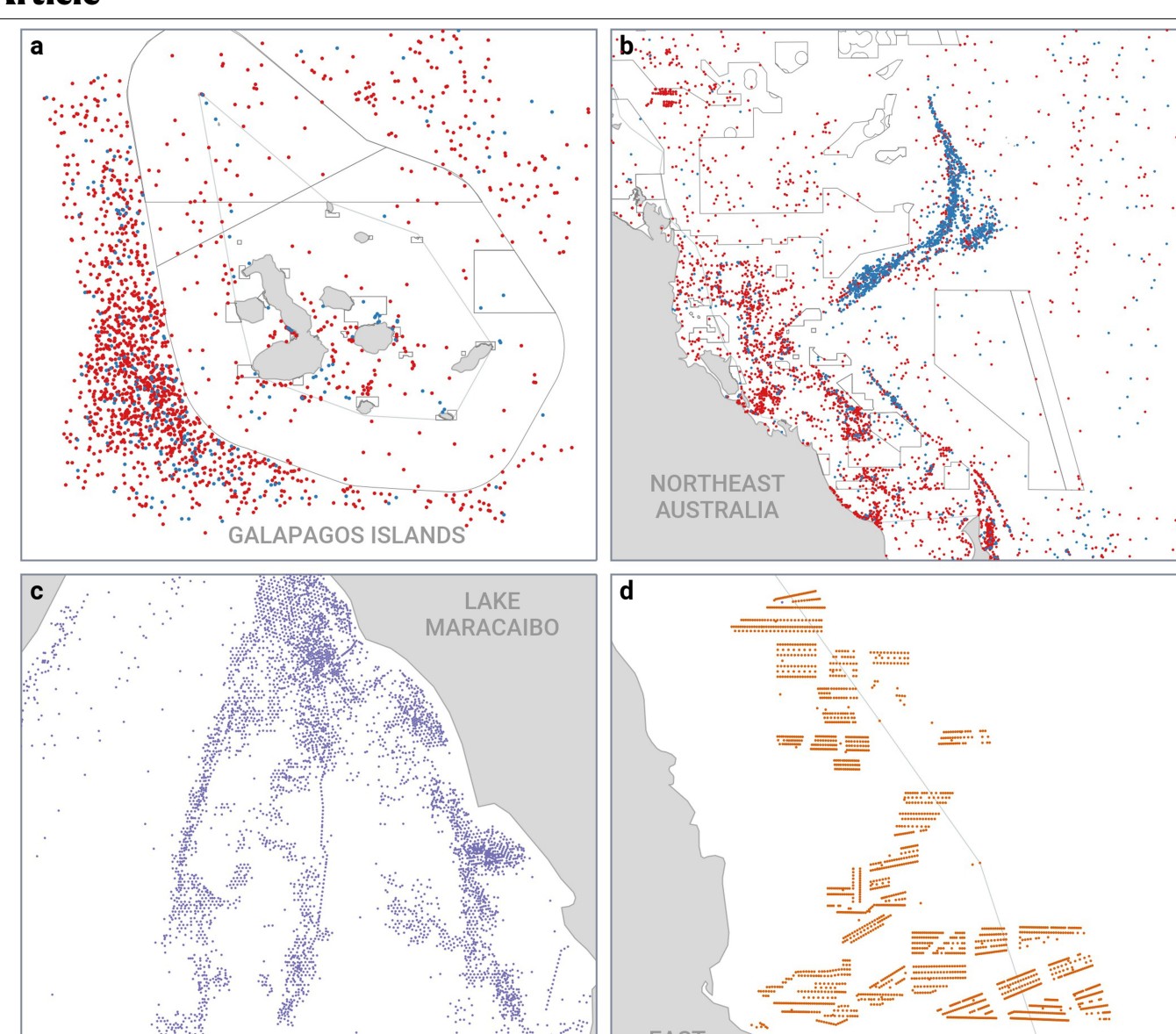

**Extended Data Fig. 7 | Satellite imagery-based detection allows monitoring at local scale. a,b**, From 2017 to 2021, there were substantial numbers of vessels not publicly tracked (red) within the boundaries of two of the most iconic, biologically important and well-monitored MPAs in the world: the Galápagos Marine Reserve and south of the Great Barrier Reef Marine Park. **c,d**, Two areas of intense marine infrastructure development are the oil infrastructure in Lake Maracaibo in Venezuela and offshore wind farms north of Shanghai, China.

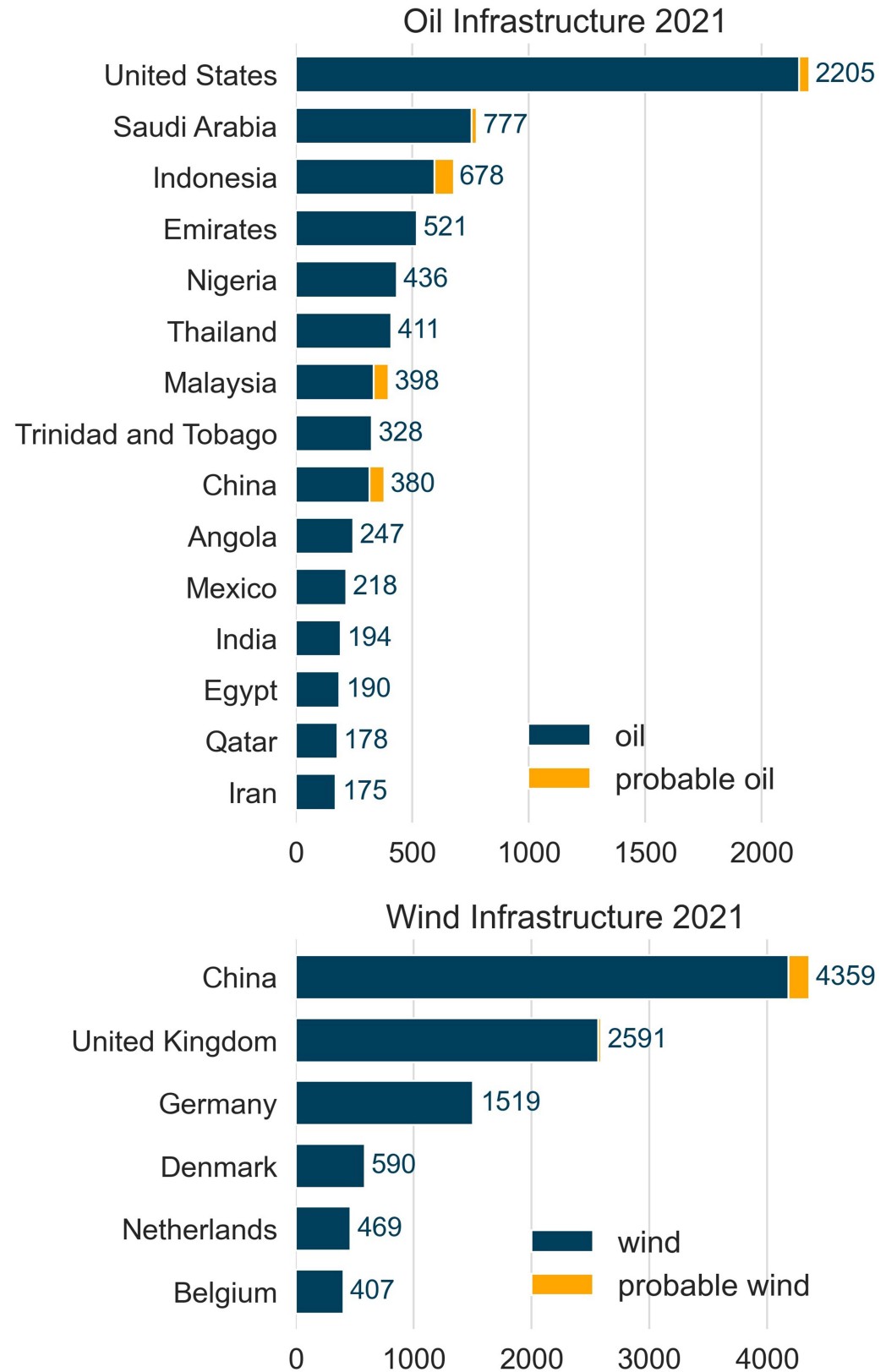

**Extended Data Fig. 8 | Leading countries with most offshore oil and wind infrastructure.** Bars represent the median value of monthly counts of offshore structures for each EEZ in 2021. 'Probable' refers to detected infrastructure with lower confidence but still within the EEZ of the respective country.

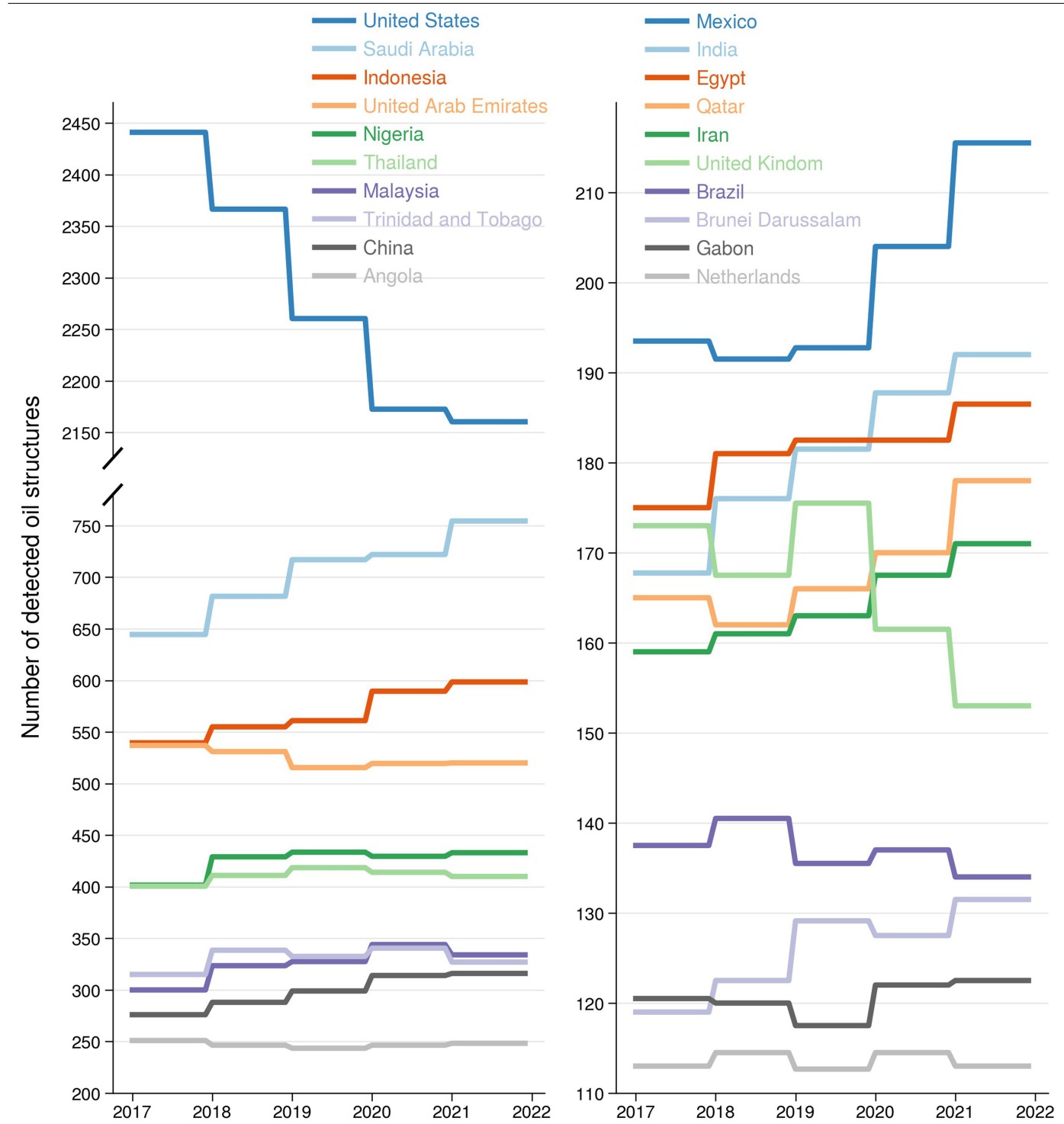

**Extended Data Fig. 9 | Offshore oil development during 2017–2021 in the top 20 oil nations.** Time series represent the median monthly counts of detected oil structures inside each country's EEZ annually. Note the different ranges in the *y* axes.

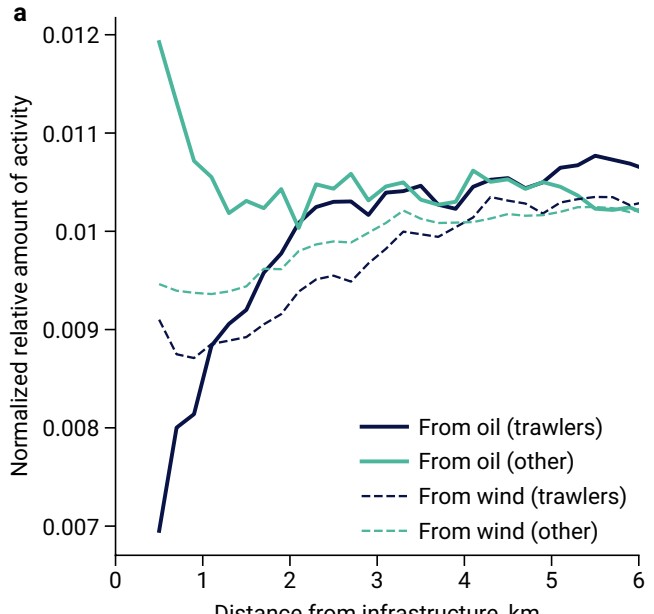

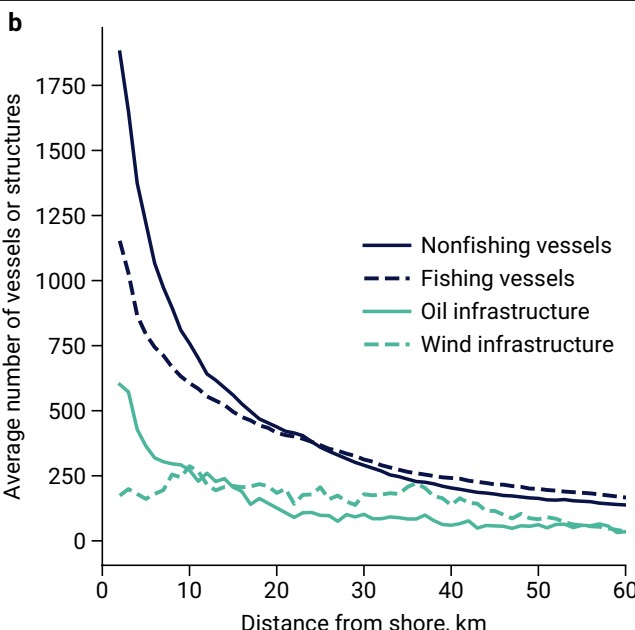

**Extended Data Fig. 10 | Number of vessels and structures as a function of distance from shore and from infrastructure. a**, Trawler vessel activity is relatively low close to oil infrastructure, but other types of fishing show increased activity there. **b**, The number of vessels and oil platforms decrease rapidly far from shore, but the number of wind structures stays relatively constant within tens of kilometres from the coast.

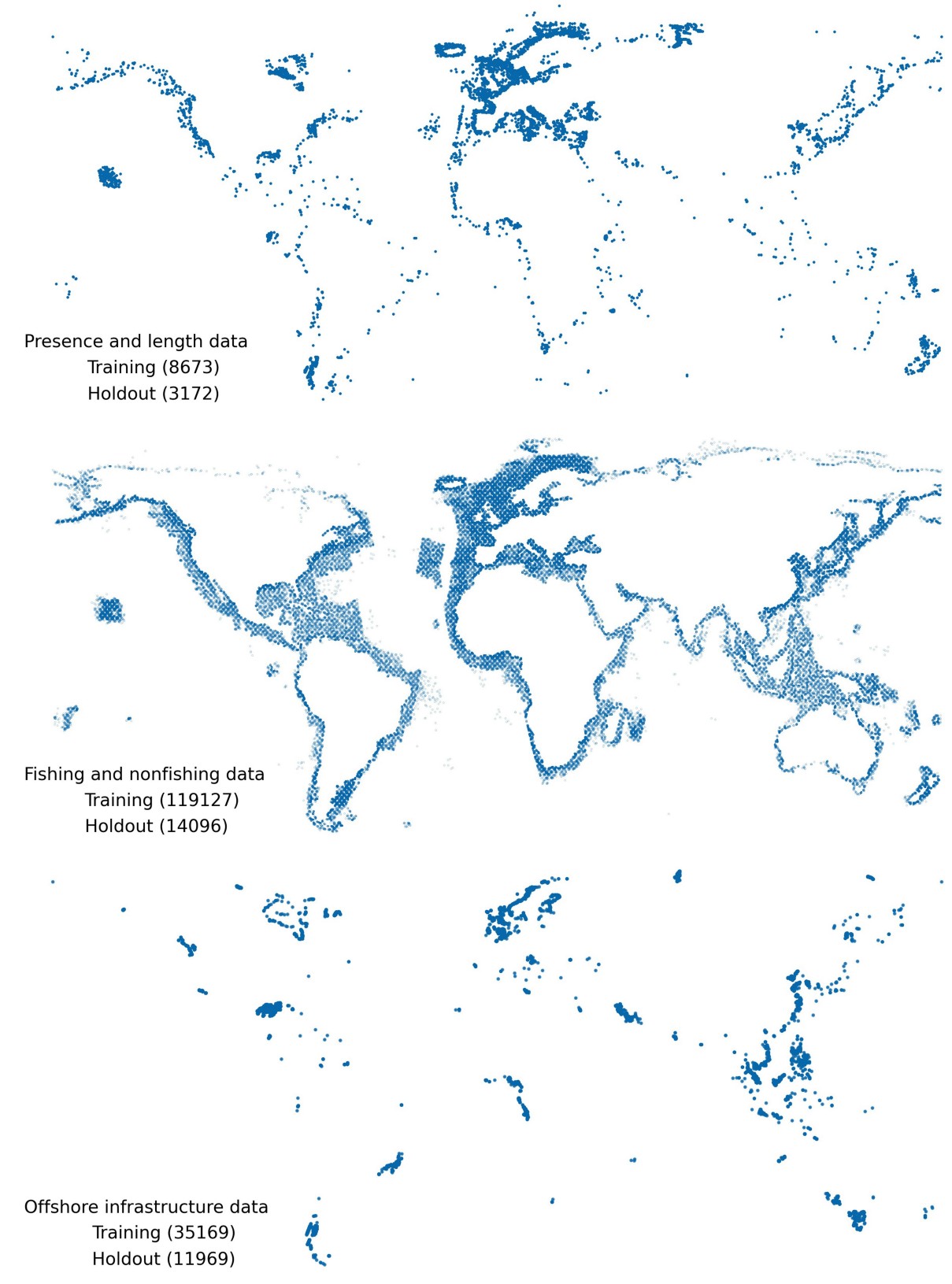

Presence and length data
Training (8673)
Holdout (3172)

Fishing and nonfishing data
Training (119127)
Holdout (14096)

Offshore infrastructure data
Training (35169)
Holdout (11969)

**Extended Data Fig. 11 | The labelled data used for training the deep-learning models sample all regions of the ocean.** Spatial distribution of the training and holdout data used to train and evaluate the 'vessel presence and length estimation' model, 'fishing and non-fishing classification' model and 'offshore infrastructure classification' model. The holdout data are random subsamples with the same spatial distribution as the training data without any overlap in time or space (no data leakage between training and test sets). See respective classification sections for a description of the sampling strategies and characteristics of each dataset.

**Extended Data Table 1 | Asia and Europe (wrongly) show comparable fishing hours if estimated from the AIS**

| Continent | Fraction of total fishing hours from AIS | Fraction of total fishing vessels from SAR |
|---|---|---|
| Asia | 0.44 | 0.71 |
| Europe | 0.36 | 0.10 |
| Africa | 0.06 | 0.07 |
| North America | 0.06 | 0.07 |
| Australia | 0.05 | 0.02 |
| South America | 0.04 | 0.04 |

Total hours of fishing activity during 2017–2021 calculated from Global Fishing Watch's AIS database, containing 52 billion vessel broadcast messages acquired from global providers ORB-COMM and Spire, compared with the fraction of total fishing vessels detected by SAR, which includes vessels not detected by AIS monitoring.