## [Peer Review File · Nature]

Manuscript Title: Satellite mapping reveals extensive industrial activity at sea

Reviewer Comments & Author Rebuttals

Reviewer Reports on the Initial Version:

Referees' comments:

Referee #1 (Remarks to the Author):

The authors of this article combine different sources of earth observation data, i.e. environmental models, satelliteborne SAR (S-1) and optical sensors (S-2), and AIS to extract an overview of worldwide shipping and offshore industry in coastal waters. Analytical methods are applied to detect maritime objects and deep learning is used to identify vessels, which are not reporting their position, and to discriminate between wind and oil related offshore infrastructures. The authors postulate that around 75% of fishing vessels are dark vessels, regularly found engaged in illegal fishing activities. Fishing activities experienced a drop of 12% during Covid-19 pandemic in 2020 to 2021, while other industrial activities remained mainly unaffected or even increased. A paradigm shift is recognized by the amount of wind related offshore infrastructures surpassing the amount of oil related offshore infrastructures during 2020.

The paper is scientifically sound and contributes to the state-of-the-art. Data and methods are made available and the derived results are comprehensible. The article deserves publication after solving the following major issue ...

Line 797-800: What are the accuracies for vessel with length over and under 50 m? According to the reference [45] (Kroodsma et al) there is a proportion of 94% of fishing vessels in the Pacific ocean and a proportion of 83% of fishing vessel in the Indian ocean. When these proportions also hold for your dataset, then the 91% accuracy of your classifier could also be achieved by simply classifying all vessels as "fishing vessel". What is the proportion of fishing vessel in the dataset of this paper?

... and the following minor issues.

Line 30+31: This sentence provides a very general statement. How is the industrial use of ocean connected to sustaining human life? You later state that fishery is even declining. Does this only hold for coastal human life, where in some regions people live from fishing? Please consider providing references for this statement. Please keep in mind that according to IPCC there is not necessarily a connection required between resources and economy or human life.

Line 52-53: Collision avoidance is not the only purpose, when AIS was introduced, see: Tetreault, B. J. "Use of the Automatic Identification System (AIS) for maritime domain awareness (MDA)", Proceedings of OCEANS 2005 MTS/IEEE, 2005

Line 306: Do you really mean absolute numbers? This absolute number then would depend strongly on the number of acquisitions per region and therefore is not expressive. In section "vessel activity estimation" you introduce average numbers for vessel activity, normalizing over time and space. Why don't you apply those average numbers here? Some more explanation is required to motivate your choice for absolute numbers, when this was your intention.

Line 560-562 and Line 987: Did you only use S-1 IW data? The specified resolution suggests that you only used IW images. The used product modes (SAR and optical products) should be mentioned at least once in your paper.

Line 642: What are the time-dependent thresholds? Which order of magnitude of false alarm rates do those thresholds imply? Please specify.

Line 714: Please clarify "vessel model". This name is not introduced. At this position of your paper, only the "vessel presence and length estimation" was described, but with ConvNeXt architecture you might rather refer to the "Fishing and non-fishing classification"

Line 744-747: What does "combined F1 score" mean? Do you mean the F1 score of the model based on SAR and optic fusion? What is the amount/proportion of classifications reached with SAR

only? What is the accuracy achieved with the SAR only model? Please specify.

Line 762-764: Does this mean you do not use any remote sensing data or image features for the classification? Can you please motivate your choice?

Line 825-828: The capabilities of your detector are in agreement with the literature on CFAR applied to SAR. However, it should be noted that the ship length is not the only parameter with influence on detectability. Especially, for smaller vessels (<25m) wind speed and incidence angle grow in importance, see: Tings, B. et al. "Modelling ship detectability depending on TerraSAR-X-derived metocean parameters" CEAS Space Journal, 2016

Line 987-994: Some of these influences have been quantified. See previous comment.

Line 1182: What do you mean by median monthly counts. There is only one bar per region, not 12. Please clarify.

Referee #2 (Remarks to the Author):

This is a truly remarkable paper destined to be an instant classic in fisheries, oil, and offshore wind development. I speak from a fisheries perspective.

This is the first effort to combine synthetic aperture radar, visual satellite images, and AIS data to identify which vessels are "dark" (with AIS turned off) in highly frequented regions of the world. The results are globally relevant, surprising, and extremely interesting.

The paper should be accepted with alacrity. Changes below are suggestions only, it could be accepted without changes.

GENERAL

1. It would be helpful on Figure 1 to show where the study area is, with an outline of regions with satellite coverage and those without. In some regions (like southern Australia) it is not clear if there are zero vessels operating, or whether that region is excluded. Similarly, there are blank regions in the East China Sea, South China Sea, south of Madagascar. Finally in the second panel it is clear that there are regions that are not considered that interrupt the plotting of shipping lanes (e.g. from South Africa to Asia).

2. Consider adding to the plot the % of vessels in each region covered by AIS. It's hard to compare this for Transport and Energy vessels with the bars being greatly different in size.

3. There is no mention in the main text of the accuracy of the AI method of detecting and classifying vessels, which is a critical part of the methods, that should be brought into the main manuscript from the Methods.

4. Line 894 "filtered out a 1-km buffer from a global shoreline". This filtering out of all vessels within 1 km of the shoreline will surely introduce some bias in the fishing vessel dataset, since a substantial number of fishing boats fish close to shore, and those close in might be more likely to have AIS turned on to avoid collisions.

MINOR COMMENTS

65 "of a couple hundred structures" is colloquial, replace with "of several hundred structures".

Fig 1 The first panel should be labeled "Industrial fishing vessels" rather than "Industrial fishing" since it tracks where vessels go, not (necessarily) where they are fishing. This is most obvious in the presence of multiple regions where the vessels are travelling along common shipping lanes in the eastern Mediterranean, from the Suez Canal to the Indian Ocean, and around southern Africa to Asia (with truncated pathways in regions with no satellite coverage).

Fig 2 panel 5 (lower right): label each line directly, and with the full country name instead of an abbreviation.

271 "number of oil and wind structures". In Fig 2D this shows an abrupt decline in the number of

oil structures in late 2019 and then a sudden increase in 2020 with the start of the Ukraine war (I assume). But this can't be the actual number of structures, this must be the number of *active* or *visited* structures. This is much more interesting but obviously needs a caveat in the text to that effect.

Figure 3: (suggestion) it would be really interesting to text-annotate these plots with interesting features, e.g. highlighting depth zones, or shelf breaks, shipping lanes, forbidden regions, marine protected areas. Does not need to be comprehensive, but would tell a more complete story to explain the kinds of easily visible features on the maps.

Figure 4 caption: previous captions refer to radar or the type of coverage, but this one just mentions "Sentinel-1". Better to say "radar from Sentinel-1".

Supp. Fig 6 and elsewhere: spell out the country names rather than using country abbreviations.

Supp. Table 1: add another column with the corrected numbers after accounting for dark vessels.

Referee #3 (Remarks to the Author):

SUMMARY:

This study uses vast amounts of satellite data (imagery and SAR) to map and catalogue sea infrastructure, both vessels and fixed infrastructure. The authors use their new dataset to highlight (1) "dark", i.e. non-tracked fishing and other vessel activity and (2) the expansion of fixed / static sea infrastructure, focused on off-shore oil, gas and wind development. Their key findings include the observation that about 75% of fishing operations are "dark" and that offshore wind infrastructure is surpassing fossil fuel developments. Incorporating a temporal aspect, the authors also analyze the effects of the COVID-19 pandemic on global fishing operations.

STRENGTHS:

This is a very ambitious project and it is clear that a lot of work went into creating the associated datasets. The analyses based on the gathered data and the conclusions drawn from them have broad implications across scientific domains. Insights from the data may be used to tackle illegal fishing operations, targeted environmental protection policies or designing food security programs. Beyond applications, the datasets can be of great interest to methodological research in remote sensing and machine learning, both of which are working broadly on developing better models for monitoring global human activity. I believe that in both dataset creation and modeling as well as analysis have been conducted with great rigor. The technical appendix is extensive and clearly outlines the steps taken.

LIMITATIONS AND AVENUES FOR IMPROVEMENT:

I am mostly very happy with this paper and only have some minor comments and questions for the authors:

- While the authors are clearly aware of spatial dependencies in their data and the (potential) problems arising from that for modeling (see especially Methods, section "Infrastructure Classification"), I feel that the spatial distribution of the training data and labels the authors use for creating their different classification and regression models could be analyzed in more detail. Specifically, could the authors write a paragraph / section about the following points: Are the labels distributed uniformly across the planet (does it make sense to provide a figure showing the distribution of training labels here?)? Following from that, I think the authors should comment on potential geographic domain shift in the data distribution and how it affects model performance in different locations: Do the trained models perform uniformly well across the globe? Are there systemic spatial variations in model accuracies? If so, can these variations be explained by e.g. the spatial distribution of the available training labels (e.g. overfitting to geographic areas with high

number of labels) or by the quality / characteristics of the satellite imagery in that area?

- I believe that there is an opportunity for strengthening the argument for the importance of this work even more: climate change. Climate change will e.g. complicate food provision to some of the poorest people on the planet, increasing their dependence on fishing. The analysis of off-shore wind farms can help inform policies around GHG emission reduction. These are just two examples
- I think a few sentences on the connection of the paper with climate change more broadly would be useful in the introduction of the paper (or wherever applicable).
- Sometimes I found it a bit hard to follow the switches between the three focus areas of the analysis portion of the paper: (1) fishing vessel detection, (2) off-shore infrastructure detection and vessels interacting with it and (3) the effects of Covid on fishing. Might it be better to separate those three analyses more clearly?
- I think it might be helpful to briefly comment on individual fisherman and small vessels which might not be covered using either tracking or satellite image recognition. Does the presence of local fishermen affect where larger vessels go? How are they affected by large, commercial (and "dark") fishing operations?
- Figure 2: Top: What are the centers of the circles? Middle/Bottom: How did they detect that a vessel interacted with fixed infrastructure?
- One of the most interesting aspects of the paper is the hand-labelled training labels (vessels, length, fishing v non-fishing). Are these going to be made available?
- For the infrastructure classification task: did the class imbalance lead to any issues during modeling?
- Where do the fishing / non-fishing training labels come from? I don't think I saw that described in the Methods section; unless I missed it, this should be added.
- Throughout the paper the authors say they analyze "2 million gigabytes" of data. Why not just say 2 petabytes?

OVERALL ASSESSMENT:

While I see some minor avenues for improving the study, I am convinced that it should be published, as its impact and relevance to a broad scientific community across domains, from ecology to remote sensing, are outstanding.

Author Rebuttals to Initial Comments:

Referee 1

Thank you for your complimentary assessment and for the list of helpful comments, which have greatly improved the paper. Below we describe our response to each concern.

Your comments are below, boxed in with a green background. Our responses follow in plain text. Any included text from the new manuscript is boxed in with white background with the line number(s) noted at the top of the quoted text.

Referee 1: Overview

The authors of this article combine different sources of earth observation data, i.e. environmental models, satellite borne SAR (S-1) and optical sensors (S-2), and AIS to extract an overview of worldwide shipping and offshore industry in coastal waters. Analytical methods are applied to detect maritime objects and deep learning is used to identify vessels, which are not reporting their position, and to discriminate between wind and oil related offshore infrastructures. The authors postulate that around 75% of fishing vessels are dark vessels, regularly found engaged in illegal fishing activities. Fishing activities experienced a drop of 12% during Covid-19 pandemic in 2020 to 2021, while other industrial activities remained mainly unaffected or even increased. A paradigm shift is recognized by the amount of wind related offshore infrastructures surpassing the amount of oil related offshore infrastructures during 2020.

The paper is scientifically sound and contributes to the state-of-the-art. Data and methods are made available and the derived results are comprehensible.

Thank you for these positive comments.

Referee 1: Major Comment 1

The article deserves publication after solving the following major issue ...

Line 797-800: What are the accuracies for vessel with length over and under 50 m? According to the reference [45] (Kroodsma et al) there is a proportion of 94% of fishing vessels in the pacific ocean and a proportion of 83% of fishing vessel in the Indian ocean. When this proportions also hold for your dataset, then the 91% accuracy of your classifier could also be achieved by simply classifying all vessels as “fishing vessel”. What is the proportion of fishing vessel in the dataset of this paper?

Kroodsma et al. (2022) focused on a part of the ocean that contains mostly fishing vessels. On a global scale, however, fishing and non-fishing vessels are roughly equal in number (see barchart in Figure 1), but they generally occupy different locations in the ocean—with fishing vessels dominating areas that are good for fishing and non-fishing vessels concentrating in shipping lanes and near ports (see, for example, Extended Data Figure 4A and 4B). As a result, a model can help us differentiate these vessels.

In our test data set, just under half of the vessels were fishing vessels. Thus, classifying all vessels as “fishing vessels” would have an accuracy of under 50%. Our model has an accuracy of 88% on vessels smaller than 50 m and 92% on vessels larger than 50 m. It is slightly less accurate on smaller vessels, as these fishing vessels sometimes share parts of the ocean with pleasure crafts or other non-fishing vessels.

We have also updated our methods to provide a sensitivity analysis of our fishing versus non-fishing vessel model, showing that the results are not very sensitive to large changes in the composition of our training data. A new paragraph in the methods describes how we performed the sensitivity analysis by calibrating our predictions with different sets of data.

Finally, we reviewed our fishing predictions region by region, consulting with regional experts at Global Fishing Watch, giving us more confidence in our general results.

Lines 632-641

“Our original *test* set contained 47% fishing and 53% non-fishing samples. We calibrated the model output scores by adjusting the ratio of fishing-to-non-fishing vessels in the test set to 1:1 [1]. We performed a sensitivity analysis to see how our results changed with different proportions of fishing and non-fishing vessels: 2:1 and 1:2. On average, about 30,000 dark vessels were detected at any given time. The calibrated scores with two thirds fishing vessels predicted that 77% of these dark vessels were fishing, while the calibration with only one third fishing predicted that 63% of the dark vessels were fishing vessels. Thus, the total percentage (considering all detections) of dark fishing and non-fishing vessels amounts to 72-76% and 21-30%, respectively. Analysts at Global Fishing Watch then reviewed these outputs in different regions of the world to verify its accuracy.”

Referee 1: Minor Comment 1

... and the following minor issues.

Line 30+31: This sentence provides a very general statement. How is the industrial use of ocean connected to sustaining human life? You later state that fishery is even declining. Does this only hold for coastal human life, where in some regions people live from fishing? Please consider providing references for this statement. Please keep in mind that according to IPCC there is not necessarily a connection required between resources and economy or human life.

Thank you for flagging this lack of clarity. We intended for this to be a topic sentence that summarizes the main points of the paragraph. However, we now see that it could also be interpreted as a stand-alone statement. We have reworded the sentence to be more precise and focus on the link between population growth and increased resource needs; we have also added several citations.

Our reasoning for the statement is as follows. A larger population will have an increased demand for energy, meaning more and further-from-shore offshore oil development and new clean energy sources (e.g. wind, waves/tides). Submarine infrastructure (e.g. cables and pipes) will need to be expanded to address new and increased demand from more countries. More people on the planet combined with a stagnant or declining

(traditional) fishing will require new/innovative industrial approaches to extracting resources from the sea (e.g. large scale aquaculture, expansion of fishing activities to more remote and challenging areas of the ocean, alternative fishing methods). Economic development will require offshore industrial exploitation of materials such as sand mining and deep sea metals; as well as an increase in the transport of goods and people across the ocean. An increase in population and expansion of ocean industries may also exacerbate territorial disputes. Finally, marine environmental degradation resulting from industrial use of the ocean directly contributes to climate change (e.g. oil spills, chemical pollution, gas emissions), the most urgent threat facing humanity. We hope that the text succinctly captures these big ideas.

Lines 29-31

“As the global population surpasses eight billion people and demand for food, energy and materials is expected to increase, industrial use of the ocean is becoming more critical to supporting modern life¹⁻³.”

Referee 1: Minor Comment 2

Line 52-53: Collision avoidance is not the only purpose, when AIS was introduced, see: Tetreault, B. J. “Use of the Automatic Identification System (AIS) for maritime domain awareness (MDA)”, Proceedings of OCEANS 2005 MTS/IEEE, 2005

Thank you for pointing out this omission, which we have corrected in the text.

Lines 52-54

“For public mapping of ships, the focus has been on the ‘automatic identification system’ (AIS)²⁴, which broadcasts vessel coordinates to track vessel movements and support maritime safety;...”

Referee 1: Minor Comment 3

Line 306: Do you really mean absolute numbers? This absolute number then would depend strongly on the number of acquisitions per region and therefore is not expressive. In section “vessel activity estimation” you introduce average numbers for vessel activity, normalizing over time and space. Why don’t you apply those average numbers here? Some more explanation is required to motivate your choice for absolute numbers, when this was your intention.

Our choice of absolute numbers in Figure 2 is intended to demonstrate the resolution of our mapping (all the way to lat-lon of individual detections) and the details of the spatial patterns that are possible with our approach (with the goal of demonstrating potential utility of our approach to future satellite missions). This figure also shows that our classification model and matching algorithm operate on individual detections (single points, not normalized grids). Note that the relative percentages of ‘dark vs. publicly tracked’ still holds, despite the differences in the number of detections per region. We present the actual quantification of “vessel activity” (requiring normalization over time and space) in Figure 1, the main figure of the paper. We have clarified our rationale in the caption for Figure 2.

Lines 343-346

“The absolute number of detections in each location depends on the local vessel density and the number of satellite image acquisitions, which varies by region. This figure shows the level of spatial detail that is possible with our mapping approach.”

Referee 1: Minor Comment 4

Line 560-562 and Line 987: Did you only use S-1 IW data? The specified resolution suggests that you only used IW images. The used product modes (SAR and optical products) should be mentioned at least once in your paper.

Correct, for Sentinel-1 (SAR) we only used the Interferometric Wide (IW) swath mode. This is the main operational mode and produces the highest resolution GRD images at ~20 m. For Sentinel-2 (optical) we used the 10 m resolution RGB and NIR bands from the Level-1C product. We have added this information to the respective sections.

Lines 397-400 (SAR)

“Our data consist of dual-polarization images (VH and VV) from the Interferometric Wide (IW) swath mode, with a resolution of about 20 m. We used the Ground Range Detected (GRD) Level-1 product provided by Google Earth Engine [3], processed for thermal noise removal, radiometric calibration and terrain correction [4].”

Lines 418-420 (Optical)

“We used the RGB and NIR bands from the Level-1C product provided by Google Earth Engine [2], and we excluded images with more than 20% cloud coverage using the QA60 bitmask band with cloud mask information.”

Referee 1: Minor Comment 5

Line 642: What are the time-dependent thresholds? Which order of magnitude of false alarm rates do those thresholds imply? Please specify.

We have specified the threshold values for the five time intervals and each satellite (S1A, S1B), as well as the detection rate for these thresholds.

Lines 476-485

“This adjustment is needed because the statistical properties of the SAR images provided by Sentinel-1 vary with time as well as by satellite (S1A and S1B); we thus found that the ocean pixels for both the mean and the standard deviation of the scenes changed, requiring different calibrations of the CFAR parameters for five different time intervals during which the statistics of the images remained relatively constant: Jan2016–Oct2016 ($n_{S1A} = 14$, $n_{S1B} = \text{None}$), Sep2016–Jan2017 (14, 18), Jan2017–Mar2018 (14, 17), Mar2018–Jan2020 (16, 19), Jan2020–Dec2021 (22, 24). The five detection thresholds were calibrated to obtain a consistent detection rate for the smaller vessels across the entire Sentinel-1 archive (60% detection of vessels 15-20 m in length).”

Referee 1: Minor Comment 6

Line 714: Please clarify “vessel model”. This name is not introduced. At this position of your paper, only the “vessel presence and length estimation” was described, but with ConvNeXt architecture you might rather refer to the “Fishing and non-fishing classification”

Here, “*vessel model*” refers to the “*vessel presence and length estimation model*”. We replaced the former with the latter in the text.

Lines 549-550

“A key difference from the ‘vessel presence and length estimation model’, besides using a different architecture, is that...”

Referee 1: Minor Comment 7

Line 744-747: What does “combined F1 score” mean? Do you mean the F1 score of the model based on SAR and optic fusion? What is the amount/proportion of classifications reached with SAR only? What is the accuracy achieved with the SAR only model? Please specify.

The F1 score presented is for the mixed-data model that uses both SAR and optical. “Combined F1 score” refers to the F1 metric for a multiclass problem, as opposed to a binary classification. In this case, we are evaluating 4 classes (oil, wind, other, noise) with a different proportion in the holdout set. We clarified this in the text. The amount of classifications done with SAR only tiles is very small, and mostly in high latitudes where there is very little infrastructure. We note that the model is the same, the only difference is that missing optical tiles are blank (we trained the model with random optical blank tiles as an augmentation strategy).

Lines 583-585

“Our best model achieved on the test set a class-weighted average F1 score of 0.99 (accuracy = 98.9%) for the multiclass problem.”

Referee 1: Minor Comment 8

Line 762-764: Does this mean you do not use any remote sensing data or image features for the classification? Can you please motive your choice?

We use a large number of features derived from remote sensing, as well as other data sets, to classify fishing versus non-fishing vessels, including: estimated vessel lengths and vessel density derived from the SAR images, hours of vessel presence derived from satellite and terrestrial AIS, environmental features such as sea surface temperature and chlorophyll derived from remote sensing, as well as geophysical information such as bathymetry. These additional data sets are described in the section “Global environmental and physical data”. We have highlighted some of the remote sensing features in the text.

Lines 608-613

“At every pixel, each raster contains contextual information on the following variables: 1) vessel density (based on SAR), 2) average vessel length (based on SAR), 3) bathymetry, 4) distance from port, 5, 6), hours of non-fishing vessel presence (from AIS) for vessels (a) under 50 m and (b) over 50 m, 7) average surface temperature, 8) average current speed, 9) standard deviation of daily temperature, 10) standard deviation of daily current speed, and 11) average chlorophyll.”

Referee 1: Minor Comment 9

Line 825-828: The capabilities of your detector are in agreement with the literature on CFAR applied to SAR. However, it should be noted that the ship length is not the only parameter with influence on detectability. Especially, for smaller vessels (<25m) wind speed and incidence angle grow in importance, see: Tings, B. et al. “Modelling ship detectability depending on TerraSAR-X-derived metocean parameters” CEAS Space Journal, 2016

Thank you for this excellent reference! We have included it in the manuscript. We also now direct the reader to “See ‘*Limitations of our study*’ for factors influencing detectability” (line 678), where we expand on these parameters.

Lines 857-866

“Vessel detection by SAR imagery is limited primarily by the resolution of the images (~20 m in the case of Sentinel-1 IW GRD products). As a result, we miss most vessels under 15 m in length, although an object smaller than a pixel can still be seen if it is a strong reflector, such as a vessel made of metal rather than wood or fiberglass. Especially for smaller vessels (< 25 m), detection also depends on wind speed and the state of the ocean⁵⁷, as a rougher sea surface will produce higher backscatter, making it difficult to separate a small target from the sea clutter. Conversely, the higher the radar incidence angle, the higher the probability of detection⁵⁷, as less backscatter from the background will be received by the antenna. The vessel orientation relative to the satellite antenna also matters, as a vessel perpendicular to the radar line of sight will have a larger backscatter cross section, increasing the probability of being detected.”

Referee 1: Minor Comment 10

Line 987-994: Some of these influences have been quantified. See previous comment.

We now make explicit reference to the above points (see our response to Comment 9 above).

Referee 1: Minor Comment 11

Line 1182: What do you mean by median monthly counts. There is only one bar per region, not 12. Please clarify.

We are referring to the median value of monthly counts (we detect infrastructure once a month). We have clarified this in the text.

Lines 1003-1004

“Bars represent the median value of monthly counts of offshore structures for each EEZ in 2021.”

Referee 2

Thank you for your complimentary assessment and for the list of helpful comments, which have greatly improved the paper. Below we describe our response to each concern.

Your comments are below, boxed in with a yellow background. Our responses follow in plain text. Any included text from the new manuscript is boxed in with white background with the line number(s) noted at the top of the quoted text.

Referee 2: Overview

This is a truly remarkable paper destined to be an instant classic in fisheries, oil, and offshore wind development. I speak from a fisheries perspective.

This is the first effort to combine synthetic aperture radar, visual satellite images, and AIS data to identify which vessels are “dark” (with AIS turned off) in highly frequented regions of the world. The results are globally relevant, surprising, and extremely interesting.

The paper should be accepted with alacrity. Changes below are suggestions only, it could be accepted without changes.

Thank you for your enthusiastic endorsement of our work. We appreciate it!

Referee 2: General Comment 1

GENERAL

1. It would be helpful on Figure 1 to show where the study area is, with an outline of regions with satellite coverage and those without. In some regions (like southern Australia) it is not clear if there are zero vessels operating, or whether that region is excluded. Similarly, there are blank regions in the East China Sea, South China Sea, south of Madagascar. Finally in the second panel it is clear that there are regions that are not considered that interrupt the plotting of shipping lanes (e.g. from South Africa to Asia).

These are great observations, and we agree that including the satellite footprint is helpful to explain the patterns you noticed. We have added the study area to Figure 1.

Referee 2: General Comment 2

2. Consider adding to the plot the % of vessels in each region covered by AIS. It's hard to compare this for Transport and Energy vessels with the bars being greatly different in size.

Great suggestion. We have added the percent of vessels publicly tracked in each continent to both maps in Figure 1.

Referee 2: General Comment 3

3. There is no mention in the main text of the accuracy of the AI method of detecting and classifying vessels, which is a critical part of the methods, that should be brought into the main manuscript from the Methods.

We agree that the performance of our models should be stated in the main text. We have added the following text:

Lines 74-77 and 79-81

“We designed and trained three deep convolutional neural networks to identify objects (>97% accuracy) and estimate their lengths ($0.84 R^2$); to classify offshore infrastructure into oil, wind, and other objects (>98% accuracy); and to classify vessels as fishing or non-fishing (>90% accuracy).” ...

“The resolution of SAR allows us to capture the majority of objects larger than 15 meters (detection rate of >70% for 25 m vessels, and >90% for 50 m and larger, Extended Data Figure 2).”

Referee 2: General Comment 4

4. Line 894 “filtered out a 1-km buffer from a global shoreline”. This filtering out of all vessels within 1 km of the shoreline will surely introduce some bias in the fishing vessel dataset, since a substantial number of fishing boats fish close to shore, and those close in might be more likely to have AIS turned on to avoid collisions.

We agree that filtering out the first kilometer from shore could introduce some bias in our fishing dataset. However, the size of this bias is likely limited because our study focuses on vessels larger than 15 m in length. These vessels tend to fish farther from shore than small scale vessels. For 15 m vessels, only 3% of AIS broadcasts occur within 1 km of shore; larger vessels spend even less time in this region.

Also, we excluded this area for a practical reason—there is much more noise from coastal features, and it is difficult to separate vessels from other features with precision in this area. We feel that the bias-variance tradeoff justifies excluding this area.

Referee 2: Minor Comment 1**MINOR COMMENTS**

65 “of a couple hundred structures” is colloquial, replace with “of several hundred structures”.

We have replaced the text as suggested.

Referee 2: Minor Comment 2

Fig 1 The first panel should be labeled “Industrial fishing vessels” rather than “Industrial fishing” since it tracks where vessels go, not (necessarily) where they are fishing. This is most obvious in the presence of multiple regions where the vessels are travelling along common shipping lanes in the eastern Mediterranean, from the Suez Canal to the Indian Ocean, and around southern Africa to Asia (with truncated pathways in regions with no satellite coverage).

Great point. We have added “vessels” to the legend of both maps.

Referee 2: Minor Comment 3

Fig 2 panel 5 (lower right): label each line directly, and with the full country name instead of an abbreviation.

We have updated Figure 4 (previously numbered Figure 2) as suggested.

Referee 2: Minor Comment 4

271 “number of oil and wind structures”. In Fig 2D this shows an abrupt decline in the number of oil structures in late 2019 and then a sudden increase in 2020 with the start of the Ukraine war (I assume). But this can't be the actual number of structures, this must be the number of *active* or *visited* structures. This is much more interesting but obviously needs a caveat in the text to that effect.

This is a great observation. The number of structures is the actual number of fixed structures, not the number visited, and we have updated the legend for Figure 4 (previously numbered Figure 2) to make this more clear. We also looked closely at the 2019-2020 transition in our infrastructure record, and the slight decline appears to be driven by a decrease in oil structures in the Gulf of Mexico. However, we note that this decline is smaller than the error bars in our oil detections. We have also slightly improved our infrastructure classification, and updated the figure and its caption accordingly.

Lines 796-801

“We then used a DBSCAN⁵⁵ clustering approach to identify detections across time (within a 50 m radius) that were likely the same structure but their coordinates differed slightly, and assigned them the most common predicted label of the cluster. We also filled in gaps for fixed structures that were missing in one timestep but detected in the previous and following timesteps, and dropped detections appearing in a single timestep.”

Referee 2: Minor Comment 5

Figure 3: (suggestion) it would be really interesting to text-annotate these plots with interesting features, e.g. highlighting depth zones, or shelf breaks, shipping lanes, forbidden regions, marine protected areas. Does not need to be comprehensive, but would tell a more complete story to explain the kinds of easily visible features on the maps.

We agree and have added annotations to Figure 2 (previously numbered as Figure 3), highlighting the following features: regional seas, shelf breaks, ocean banks, submarine channel, canyon system, aquaculture farms, and country names.

Referee 2: Minor Comment 5

Figure 4 caption: previous captions refer to radar or the type of coverage, but this one just mentions “Sentinel-1”. Better to say “radar from Sentinel-1”.

We have made the suggested change to Figure 3 (previously numbered as Figure 4). Good eye!

Referee 2: Minor Comment 6

Supp. Fig 6 and elsewhere: spell out the country names rather than using country abbreviations.

We now spell out the country names in all figures.

Referee 2: Minor Comment 7

Supp. Table 1: add another column with the corrected numbers after accounting for dark vessels.

We have added the following column to Table 1: “Fraction of total fishing vessels from SAR.”

Referee 3

Thank you for your complimentary assessment and for the list of helpful comments, which have greatly improved the paper. Below we describe our response to each concern.

Your comments are below, boxed in with a blue background. Our responses follow in plain text. Any included text from the new manuscript is boxed in with white background with the line number(s) noted at the top of the quoted text.

Referee 1: Summary and Strengths

SUMMARY:

This study uses vast amounts of satellite data (imagery and SAR) to map and catalogue sea infrastructure, both vessels and fixed infrastructure. The authors use their new dataset to highlight (1) “dark”, i.e. non-tracked fishing and other vessel activity and (2) the expansion of fixed / static sea infrastructure, focused on off-shore oil, gas and wind development. Their key findings include the observation that about 75% of fishing operations are “dark” and that offshore wind infrastructure is surpassing fossil fuel developments. Incorporating a temporal aspect, the authors also analyze the effects of the COVID-19 pandemic on global fishing operations.

STRENGTHS:

This is a very ambitious project and it is clear that a lot of work went into creating the associated datasets. The analyses based on the gathered data and the conclusions drawn from them have broad implications across scientific domains. Insights from the data may be used to tackle illegal fishing operations, targeted environmental protection policies or designing food security programs. Beyond applications, the datasets can be of great interest to methodological research in remote sensing and machine learning, both of which are working broadly on developing better models for monitoring global human activity. I believe that in both dataset creation and modeling as well as analysis have been conducted with great rigor. The technical appendix is extensive and clearly outlines the steps taken.

Thank you for these positive comments on our paper’s quality and contribution.

Referee 3: Minor Comment 1

LIMITATIONS AND AVENUES FOR IMPROVEMENT:

I am mostly very happy with this paper and only have some minor comments and questions for the authors:

- While the authors are clearly aware of spatial dependencies in their data and the (potential) problems arising from that for modeling (see especially Methods, section “Infrastructure Classification”), I feel that the spatial distribution of the training data and labels the authors use for creating their different classification and regression models could be analyzed in more detail. Specifically, could the authors write a paragraph / section about the following points: Are the labels distributed uniformly across the planet

(does it make sense to provide a figure showing the distribution of training labels here?)? Following from that, I think the authors should comment on potential geographic domain shift in the data distribution and how it affects model performance in different locations: Do the trained models perform uniformly well across the globe? Are there systemic spatial variations in model accuracies? If so, can these variations be explained by e.g. the spatial distribution of the available training labels (e.g. overfitting to geographic areas with high number of labels) or by the quality / characteristics of the satellite imagery in that area?

We agree that we should expand on the creation of our training datasets. Good training data is the most important factor in any machine learning problem, and we appreciate the opportunity to clarify our methods here. We experimented extensively with different training-validation-test setups, sampling strategies, and label distributions to find the “optimal” training dataset for each problem. As you correctly point out, a key aspect of a training dataset is capturing the wide range of examples across the ocean, without emphasizing any particular region. Our training sets, therefore, contain samples from every area of the ocean imaged by Sentinel-1 while keeping the number of samples per region consistent. To achieve this consistency across regions we thinned the data in regions with substantially more samples, e.g. around European waters. We also note that there is no spatial leakage (overlap) between the training and test sets at any point in time. Given these characteristics in the training and test data we expect the overall model performance to be consistent across different regions. There are, however, some corner cases at local scales for which the model might perform better or worse than the average (we describe some of these cases in the “Limitations of our study”). We have added to the Methods an additional figure (Extended Data Figure 11, reproduced below), which shows the spatial distribution of our training datasets and highlights in the caption the key points raised here.

Lines 1016-1023

“Extended Data Figure 11 | The labeled data used for training the deep learning models sample all regions of the ocean. (from top to bottom) Spatial distribution of the training and holdout data used to train and evaluate the ‘vessel presence and length estimation model’, ‘fishing and non-fishing classification model’, and ‘offshore infrastructure classification model’. The holdout data are random subsamples with the same spatial distribution as the training data without any overlap in time or space (no data leakage between training and test sets). See respective classification sections for a description of the sampling strategies and characteristics of each data set.”

Referee 3: Minor Comment 2

- I believe that there is an opportunity for strengthening the argument for the importance of this work even more: climate change. Climate change will e.g. complicate food provision to some of the poorest people on the planet, increasing their dependence on fishing. The analysis of off-shore wind farms can help inform policies around GHG emission reduction. These are just two examples – I think a few sentences on the connection of the paper with climate change more broadly would be useful in the introduction of the paper (or wherever applicable).

We agree that the connection of our study with climate change is a key argument worth mentioning. The fishing dynamics and the growing industrialization highlighted in this paper, together with climate change, are the three major forces that are changing the ocean. We added a sentence in the conclusion making this connection.

Lines 205-207

“Our data can also help quantify the true scale of greenhouse gas (GHG) emissions from vessel traffic and offshore development, informing policies on reducing GHG emissions.”

Referee 3: Minor Comment 3

- Sometimes I found it a bit hard to follow the switches between the three focus areas of the analysis portion of the paper: (1) fishing vessel detection, (2) off-shore infrastructure detection and vessels interacting with it and (3) the effects of Covid on fishing. Might it be better to separate those three analyses more clearly?

Thank you for flagging this important issue. We have added subheadings and reorganized the text to delineate the main focus areas. We think this has greatly improved the flow of the paper.

Referee 3: Minor Comment 4

- I think it might be helpful to briefly comment on individual fisherman and small vessels which might not be covered using either tracking or satellite image recognition. Does the presence of local fishermen affect where larger vessels go? How are they affected by large, commercial (and “dark”) fishing operations?

The interactions between industrial fishing vessels and small scale vessels are indeed an important potential application of this dataset. For example, a number of countries have established “preferential access areas” (PAAs) where small scale fishers have preferential access over industrial fishers. Our dataset could help identify where larger vessels interact with these smaller vessels. We hope that the sentence in the second to last paragraph, which mentions artisanal fishing, covers this idea. It is beyond the scope of our study to say more about these interactions, but we expect our data to facilitate future research on this important topic.

Lines 200-203

“With our freely available dataset and technology, hotspots of potentially illegal activity can now be revealed²⁸ and industrial fishing vessels can be identified that are encroaching on artisanal fishing grounds¹⁷ or other countries’ EEZs²⁷, but at a global scale and accessible to any nation.”

Referee 3: Minor Comment 5

- Figure 2: Top: What are the centers of the circles? Middle/Bottom: How did they detect that a vessel interacted with fixed infrastructure?

These are important points to clarify. Thanks for pointing this out. For the top panel, each circle is proportional to the number of structures in each 100 km² grid cell (at the center of the circle). For the middle/bottom panels, we filtered all vessels in our AIS database that were within 100 m of an oil or wind structure at some point during 2021. We clarified both of these details in the caption of Figure 4 (previously numbered as Figure 2).

Lines 363-375

“**Figure 4** | The growing footprint of offshore development extends far beyond the fixed infrastructure. (top) Global map of offshore development, showing oil infrastructure in major oil-producing areas, wind farms, and other human-made structures (such as piers, power lines, and aquaculture). Circles are proportional to the number of structures per 100 km² grid cells. (middle) One year of vessel traffic associated with offshore infrastructure in the North Sea. These vessels were all broadcasting and interacted with (were closer than 100 m from) a detected oil or wind infrastructure at some point during 2021. Globally, in 2021, nearly 4,140,000 hours of vessel activity were associated with oil platforms, and around 792,500 hours with wind turbines. (bottom) Evolution of the number of (fixed) oil and wind structures in the ocean (left), and the leading nations in wind development (right). Extended Data Figure 9 shows the leading nations in oil development. Error bars define a lower bound considering only high confidence detections of oil structures, and an upper bound including detections with lower confidence (e.g. potential oil structures outside oil-producing areas; see Methods).”

Referee 3: Minor Comment 6

- One of the most interesting aspects of the paper is the hand-labelled training labels (vessels, length, fishing v non-fishing). Are these going to be made available?

Yes, all data related to this study will be made freely available through the Global Fishing Watch data portal and/or the study’s replication page on Zenodo. We have specified how to access these data in the ‘*Data availability*’ statement.

Lines 896-899

“All vessel and infrastructure data are freely available through the Global Fishing Watch data portal at <https://globalfishingwatch.org>. All data to reproduce this study can be downloaded from <https://doi.org/10.5281/zenodo.8256932> (statistical analyses and figures) and <https://doi.org/10.5281/zenodo.8258124> (model training and evaluation).”

Referee 3: Minor Comment 7

- For the infrastructure classification task: did the class imbalance lead to any issues during modeling?

We found no issues related to the class imbalance in the fixed infrastructure classification. This “imbalance” reflects the actual distribution of fixed infrastructure, with oil-related structures spread across the ocean and wind-related structures mostly confined to two regions, the North Sea and Southeast China, meaning there is not much variability in wind turbines samples. Furthermore, wind turbines appearing surrounded by neighboring turbines are very distinct compared to oil platforms and other structures like fish cages. In this case, the model can “learn” to differentiate between these structures with relatively few samples. The new Extended Data Figure 11, reproduced above, shows the distribution of the training data.

Referee 3: Minor Comment 8

- Where do the fishing / non-fishing training labels come from? I don't think I saw that described in the Methods section; unless I missed it, this should be added.

Thanks for raising this issue; we have clarified this section of the text. This information is from AIS data.

Lines 616

“We obtained the fishing and non-fishing labels from AIS vessel identities²⁶.”

Referee 3: Minor Comment 9

- Throughout the paper the authors say they analyze “2 million gigabytes” of data. Why not just say 2 petabytes?

Good point. We now use “petabytes” throughout the paper.

Referee 3: Overall Assessment

OVERALL ASSESSMENT:

While I see some minor avenues for improving the study, I am convinced that it should be published, as its impact and relevance to a broad scientific community across domains, from ecology to remote sensing, are outstanding.

Thank you. We greatly appreciate your endorsement, as well as your constructive feedback on ways to improve the paper.

Reviewer Reports on the First Revision:

Referees' comments:

Referee #1 (Remarks to the Author):

The authors did answer my previous comments and much improvement is noted, though the first version of the paper was already in a good state. Again, the paper deserves publication in the journal.

I have some minor comments:

Still, I am missing the answer to one of my previous comments ("Referee 1: Minor Comment 5") about the actual constant false alarm rate implied by the set time-dependent CFAR thresholds (which are now provided):

Can you please give some insight to the resulting order of magnitude of false alarm rates from the set thresholds? In section "False positives and recall" in line 664 you calculated a rate of 2% using an independent method, which could be considered as an actual false alarm rate taking also artefacts etc. into account (also see next comment). From your high thresholds I would however assume a much lower theoretical level of false alarm rate, which would be calculated from the gaussian distributions of background pixel values defined by your μ_b and σ_b and the x_{px} (or n_t) separating the distribution into proportion of pixels values labelled as part of the background (true negative) and labelled as target (false positive (or false alarm)).

Lines 774 to 787: Please give more information and clarification on the identification of Azimuth ghost signals. I cannot follow the current description: How was the Azimuth ambiguity distance calculated? Did you also take the small (perhaps negligible) shift in range direction into account (see for example "Azimuth Ambiguities Removal in Littoral Zones Based on Multi-Temporal SAR Images" by Xiangguang Leng, Kefeng Ji, Shilin Zhou, and Huanxin Zou published 2017 in Remote Sensing)? Did you discard all detections within 200 m above and below the actual detection in Azimuth direction? What do you mean by (line 783) "[...] detected objects larger than 100 m at a few kilometers away from the satellite [...]" (all detections should be several hundreds of kilometers away from the satellites!)? For vessels above 100 m I would expect a larger percentage of occurring Azimuth ambiguities than 0.6%, which then would significantly reduce the number of dark non-fishing vessels (so not really changing the essential statements of your study, though). I think the paragraph deserves some revision and correction.

Figure 4, Extended Data Figure 7: Geographical locations/names are appreciated also here, as already given for other figures containing maps.

Extended Data Figure 6: Please highlight the periods of Chinese fishing moratorium, as already done in Figure 3

Repeatedly: font changes after Figure references, e.g. in lines 168-169 or 172

Referee #3 (Remarks to the Author):

I want to thank the authors for their thorough reply to all reviewer comments. My (minor) concerns are all addressed and / or I am satisfied with the authors' justification of their choices in presenting the results of the paper. I am particularly happy about the Extended Data Figure 11.

At this stage, I believe that this paper is ready for publication, and I recommend acceptance. Congratulations to the authors on this outstanding work!

Author Rebuttals to First Revision:

Referee 1

Referee 1: Overview

The authors did answer my previous comments and much improvement is noted, though the first version of the paper was already in a good state. Again, the paper deserves publication in the journal.

Thank you! Please find below our responses to your remaining comments.

Referee 1: Minor Comment 1

Still, I am missing the answer to one of my previous comments (“Referee 1: Minor Comment 5”) about the actual constant false alarm rate implied by the set time-dependent CFAR thresholds (which are now provided):

Can you please give some insight to the resulting order of magnitude of false alarm rates from the set thresholds? In section “False positives and recall” in line 664 you calculated a rate of 2% using an independent method, which could be considered as an actual false alarm rate taking also artefacts etc. into account (also see next comment). From your high thresholds I would however assume a much lower theoretical level of false alarm rate, which would be calculated from the gaussian distributions of background pixel values defined by your μ_b and σ_b and the x_{px} (or n_t) separating the distribution into proportion of pixels values labelled as part of the background (true negative) and labelled as target (false positive (or false alarm)).

The reviewer makes a good point about the theoretical false alarm rate based on thresholds and gaussian distributions of the background pixel values. We do not have these theoretical estimates, however, because our CFAR approach is empirical. We calibrated our CFAR parameters empirically to obtain a consistent detection rate across five years of imagery, in which the quality and, therefore, the statistics of the SAR images varies. This variation occurred in discrete time intervals likely related to changes in the image formation procedure (e.g. processing/correction improvements of the raw imagery). In this scenario, a theoretical CFAR has two key limitations. First, it assumes a statistical model of the background (e.g. a two-parameter gamma-distribution or three-parameter K -distribution) and this statistical model is assumed to be the same for all images. Second, closed-form estimators for the distribution parameters and theoretical thresholds are derived. These operations are complex and usually computationally expensive. In large-scale studies, such as ours with over a petabyte of imagery, this approach is not feasible and often suboptimal (see Crisp 2004 [1] for an extensive discussion of different CFAR approaches). Thus, the theoretical false alarm rate is not relevant in this context.

Instead, as the reviewer correctly points out, we prefer to assess the *actual* false positive rate independently of any assumptions of the CFAR model, and also accounting for other factors outside of the detection model. Our false alarm rate is, therefore, empirical and requires aggregating results over different geographic areas, multiple time intervals, and from different analysts. Our estimate is also an upper bound,

as it assumes that all the detections in these regions are false positives where, in fact, some may be dark vessels. The actual false positive rate is likely lower, precisely as the reviewer suggests.

[1] Crisp, D.J., The State-of-the-Art in Ship Detection in Synthetic Aperture Radar Imagery (2004).

Referee 1: Minor Comment 2

Lines 774 to 787: Please give more information and clarification on the identification of Azimuth ghost signals. I cannot follow the current description: How was the Azimuth ambiguity distance calculated? Did you also take the small (perhaps negligible) shift in range direction into account (see for example “Azimuth Ambiguities Removal in Littoral Zones Based on Multi-Temporal SAR Images” by Xiangguang Leng, Kefeng Ji, Shilin Zhou, and Huanxin Zou published 2017 in Remote Sensing)? Did you discard all detections within 200 m above and below the actual detection in Azimuth direction? What do you mean by (line 783) “[...] detected objects larger than 100 m at a few kilometers away from the satellite [...]” (all detections should be several hundreds of kilometers away from the satellites!)? For vessels above 100 m I would expect a larger percentage of occurring Azimuth ambiguities than 0.6%, which then would significantly reduce the number of dark non-fishing vessels (so not really changing the essential statements of your study, though). I think the paragraph deserves some revision and correction.

We agree that this description could have been clearer. We have revised and expanded the text to clarify that our analysis approach robustly accounts for radar ambiguities. As we note in the Methods, ambiguities affect a very small fraction of detections overall, and, as the reviewer points out, this will not affect our findings.

We took this opportunity to revise our method and make it simpler by using the angle from the detection to the satellite’s nadir (off-nadir angle) and from the detection to the ambiguity (azimuth angle). Using angles directly instead of distances in km allows locating the ambiguities more precisely. Our revised estimate is that 0.51% (previously 0.60%) of detections are ambiguities. All the final numbers reported in this paper remain unchanged.

This figure we report refers to the number of detections that are likely to be ambiguities resulting from vessels in the ocean. Our analysis suggests that only objects larger than ~130 meters (as estimated from SAR) regularly present ambiguities that result in detections. And vessels larger than this size constitute only ~20% of all detections – which implies that the rate of ambiguities from large vessels is substantially higher than half a percent. We note that the number we present refers to ambiguities that are detected by our CFAR algorithm *and* not removed by our neural net classification, which eliminates most of the noise (the training data included ambiguities).

Small shifts in the range direction, as the reviewer notes, appear to be negligible. We calculated the line in the azimuth direction for vessel detections larger than 100 m and

identified detections within 200 meters of this line. The 200 meter figure was determined empirically – we found that almost all potential ambiguities were within this distance from the line.

We can inspect some of our figures to verify that we have eliminated most azimuth ambiguities. Azimuth ambiguities from vessels show up as “ghost” detections offset a few kilometers (to the north or south because the satellite travels north-south) from the original targets. These azimuth ambiguities would produce ghost shipping lanes of smaller vessels parallel to east-west shipping lanes. Figure 2c shows the strait of Gibraltar, through which passes a major shipping lane, traveling east-west. If we were incorrectly detecting ambiguities from large shipping vessels and classifying them as fishing, they would be visible in this figure as a line of small, dark vessels parallel to the shipping lane.

We agree that we were unclear in our use of “distance from satellite,” as the reviewer highlighted. What we actually calculated was the distance from the detections to the satellite ground track. Because the satellite’s altitude does not vary much, this distance is a reasonable proxy for the distance to the satellite, as well as for the satellite’s view angle (see our updated description below).

Ambiguities that are created by objects on shore or by offshore infrastructure were already excluded from this analysis because we removed repeated (fixed) objects from the vessel detection dataset, exactly as it is recommended by the reference shared by the reviewer.

Regarding the infrastructure detection dataset, our offshore infrastructure classification model can differentiate between ambiguities and true fixed infrastructure because, in addition to Sentinel-1 SAR, it also uses optical imagery from Sentinel-2, which is free from radar ambiguities. We explain our deep learning classification in the respective section of the Methods. In the training data for this model, we included several thousand examples of ambiguities, which we identified by looking for repeated objects that appeared only in an ascending or descending satellite path. While fixed infrastructure appears in images from both ascending and descending passes, ambiguities do not as they depend on the satellite scan direction, so they will appear as fixed objects only on ascending or descending imagery, not both.

We have expanded the respective section in the Methods to describe our treatment of ambiguities.

Lines 770-804

A minor source of false positives is “radar ambiguities” or “ghosts”, which are an aliasing effect caused by the periodic sampling (radar echoes) of the target to form an image. For Sentinel-1, these ghosts are most commonly caused by bright objects and appear offset a few kilometers in the azimuth direction (parallel to the satellite ground track) from the source object. These ambiguities appear separated from their source by an azimuth angle⁵⁵ $\psi = \lambda / (2V) PRF$, where λ is the SAR wavelength, V is the satellite velocity, and PRF is the SAR pulse repetition frequency, which in the case of Sentinel-1 ranges from 1-3 kHz and is constant across each sub-swath of the image³⁵. Thus we expect the offsets to also be constant across each sub-swath.

To locate potential ambiguities, we calculated the off-nadir angle³⁵ θ_i for every detection i and then identified all detections j within 200 m of the azimuth line through each detection as candidate ambiguities. We then calculated the difference in azimuth angles ψ_{ij} for these candidates. To find which of these detections were potential ambiguities, we binned the calculated off-nadir angles (θ_i) in intervals of 0.1 degrees (approximately 200 m), and built a histogram for each interval by counting the number of detections at different azimuthal offset angles ψ , binning ψ at 0.001 degrees. For each interval θ_i , we identified the angle ψ where there was the maximum number of detections, limiting ourselves to cases where the number of detections was at least two standard deviations above the background level. As expected, ambiguities appeared at a consistent ψ within each of the three sub-swaths of IW mode images. For $\theta < 32.41$ degrees, ambiguities occurred at $\psi = 0.363 \pm 0.004$ degrees. For $32.41 < \theta < 36.87$, ambiguities occurred at $\psi = 0.308 \pm 0.004$ degrees. And for $\theta > 36.85$ degrees, ambiguities occurred at $\psi = 0.359 \pm 0.004$ degrees.

We then flagged all pairs of detections that lay along a line parallel to the satellite ground track and had an angle ψ within the expected values for their respective sub-swath. The smaller (dimmer) object in the pair was then selected as a potential ambiguity. We identified about 120,000 outliers out of 23.1 million detections (0.5%), which we excluded from our analysis.

Ambiguities can also arise from objects on shore. Because, generally, only objects larger than 100 m produce ambiguities in our data, and few objects larger than 100 m on shore regularly move, these ambiguities are likely to show up in the same location in images at different times. All stationary objects were removed from our analysis of vessels. The analysis of infrastructure also removed these false detections because, in addition to SAR, it draws on Sentinel-2 optical imagery, which is free from these ambiguities.

References:

35. https://sentinel.esa.int/documents/247904/349449/s1_sp-1322_1.pdf

55. <https://www.mdpi.com/2072-4292/13/23/4865>

Referee 1: Minor Comment 3

Figure 4, Extended Data Figure 7: Geographical locations/names are appreciated also here, as already given for other figures containing maps.

We have added country names to these figures.

Referee 1: Minor Comment 4

Extended Data Figure 6: Please highlight the periods of Chinese fishing moratorium, as already done in Figure 3

We have highlighted the periods of Chinese moratoria in the figure.

Referee 1: Minor Comment 5

Repeatedly: font changes after Figure references, e.g. in lines 168-169 or 172

Thanks for catching this! We have fixed the formatting.

Reviewer Reports on the Second Revision:

Referee #1

Thank you for clarifications and adaptations following my previous minor comments. I have no further comments and consider the paper ready for publication.